# Autoformulation of Mathematical Optimization Models Using LLMs

## Abstract

Mathematical optimization is fundamental to decision-making across diverse domains, from operations research to healthcare. Yet, translating real-world problems into optimization models remains a formidable challenge, often demanding specialized expertise. This paper formally introduces the concept of *autoformulation*—an automated approach to creating optimization models from natural language descriptions for commercial solvers. We identify the three core challenges of autoformulation: (1) defining the vast, problem-dependent hypothesis space, (2) efficiently searching this space under uncertainty, and (3) evaluating formulation correctness (ensuring a formulation accurately represents the problem). To address these challenges, we introduce a novel method leveraging *Large Language Models* (LLMs) within a *Monte-Carlo Tree Search* framework. This approach systematically explores the space of possible formulations by exploiting the hierarchical nature of optimization modeling. LLMs serve two key roles: as dynamic formulation hypothesis generators and as evaluators of formulation correctness. To enhance search efficiency, we introduce a pruning technique to remove trivially equivalent formulations. Empirical evaluations across benchmarks containing linear and mixed-integer programming problems demonstrate our method's superior performance. Additionally, we observe significant efficiency gains from employing LLMs for correctness evaluation and from our pruning techniques.

## 1 Introduction

Mathematical optimization has long been a cornerstone of decision-making processes across various domains, from supply chain management (Bramel & Simchi-Levi, 1997) and healthcare resource allocation (Delgado et al., 2022) to portfolio optimization (Mokhtar et al., 2014). These problems are characterized by maximizing an objective function subject to constraints (Williams, 2013). Traditionally, optimization modeling follows a three-step process: ▶ gathering problem requirements, typically expressed in unstructured formats and domain terminology; ▶ formulating these requirements into a formal mathematical model, including variables, constraints, and objective functions; ▶ implementing the model computationally using specialized modeling language for solution using commercial solvers. These solvers (e.g. Gurobi (Gurobi Optimization, LLC, 2024), CPLEX (Cplex, 2009)), leverage sophisticated algorithms to tractably optimize a wide array of problems to *global* optimality, including convex problems, (e.g. linear and quadratic programs), and certain non-convex problems (e.g. mixed-integer linear programs) (Boyd & Vandenberghe, 2004).

**Autoformulation.** Despite major advances in solving algorithms over the past decades, the process of formulating optimization models still relies largely on human expertise (Karmarkar, 1984). Autoformulation aims to address this bottleneck by automating the formulation process, bridging the gap between problem descriptions and formal mathematical models. This approach enhances time and cost efficiency while enabling access for users without deep optimization expertise. At its core, autoformulation can be conceptualized as a search for an optimal formulation within a vast hypothesis space. However, this process is complicated by several challenges. For one, the hypothesis space is highly problem-dependent, making it difficult to manually specify. Second, efficiently searching through this hypothesis space requires methods that can balance exploitation and exploration, particularly given the uncertainty in the optimal formulation. Lastly, the search should be guided by a signal of formulation correctness, which is itself an ill-posed problem. While solvers can indi-

cate optimality gap and computational efficiency, evaluating that the formulated model accurately represents the intended real-world problem remains a distinct challenge.

A keen insight of this work is that recent advances in *Large Language Models* (LLMs) (Brown, 2020; Chowdhery et al., 2023) have opened new avenues for autoformulation. LLMs contribute several crucial capabilities to this process: contextual understanding for nuanced interpretation of problem descriptions, vast domain knowledge to incorporate relevant modeling techniques, and reasoning capabilities to support approximate evaluation of formulation correctness. Recent works (Ramamonjison et al., 2023; Xiao et al., 2023; AhmadiTeshnizi et al., 2024) have demonstrated the promising potential of LLMs in autoformulation, laying important groundwork in this field. Building upon these contributions, our work focuses on developing techniques for efficient, systematic exploration and introducing novel methods for evaluating formulation correctness.

**Key considerations.** We conceptualize autoformulation as a search problem, leveraging optimization modeling's inherent hierarchical structure to efficiently explore the vast hypothesis space, guided by feedback on formulation correctness. We decompose optimization modeling into hierarchical components and introduce a *Monte-Carlo Tree Search* (MCTS) method to incrementally explore each component's formulation space (Coulom, 2006). We employ LLMs for two specialized roles: (1) as context-dependent *hypothesis generators* to produce component formulations at each level of the tree search; (2) as *evaluators* of formulation correctness, which is combined with solver-returned data to obtain a reward signal to guide exploration. To further improve search efficiency, we introduce a pruning technique using *Satisfiability Modulo Theories* solvers, eliminating redundant hypotheses that are syntactically different yet functionally equivalent (Barrett & Tinelli, 2018). Empirically, we observed that this significantly reduces search efforts expended on equivalent formulations. Our algorithm systematically explores the hypothesis space through multiple iterations, producing a set of functionally distinct models, each scored using our evaluation function.

**Contributions.** Our main contributions are: A①We formally introduce *autoformulation* of mathematical optimization models, framing it as a search problem and identifying its core challenges. ② We propose a novel approach integrating LLMs as hypothesis generators and evaluators within an MCTS framework, enabling efficient systematic exploration of the optimization model space. ③ Across two benchmarks containing linear and mixed-integer programming problems, we demonstrate our method's superior performance in formulating correct models, observing efficiency gains from pruning and LLM evaluation of formulation correctness.

## 2 AUTOFORMULATION: TOWARDS AUTOMATED OPTIMIZATION MODELING

Optimization modeling seeks to minimize an objective function subject to specific constraints on decision variables (Dantzig, 1990). The mathematical model can be expressed in a **general form**:

$$
\begin{aligned}
\text{Minimize} \quad & f(\mathbf{x}) \\
\text{subject to} \quad & g_i(\mathbf{x}) \leq 0, \quad i = 1, \ldots, I, \\
& h_j(\mathbf{x}) = 0, \quad j = 1, \ldots, J.
\end{aligned}
\tag{1}
$$

Here $\mathbf{x} \in \mathcal{X}$ represents the vector of decision variables, and $\mathcal{X} \subseteq \mathbb{R}^\ell \times \mathbb{Z}^k$ is the domain of the problem for which the objective and constraints functions are all defined. Furthermore, $f : \mathcal{X} \to \mathbb{R}$ is the objective function to be minimized, $g_i : \mathcal{X} \to \mathbb{R}$ are inequality constraints, $h_j : \mathcal{X} \to \mathbb{R}$ are equality constraints, and $I$ and $J$ are the numbers of inequality and equality constraints respectively. The feasible region is the set of all possible points that satisfy the problem constraints: $\{\mathbf{x} \in \mathcal{X} \mid g_i(\mathbf{x}) \leq 0, \ \forall i \in [I], h_j(\mathbf{x}) = 0, \ \forall j \in [J]\}$.

**Convex problems.** An optimization problem is **convex** if $f$ and $g_i \ \forall i \in [I]$ are convex, and $h_j \ \forall j \in [J]$ are affine. Convexity is significant as any local optimum of a convex problem is globally optimal, and specialized solvers can efficiently solve convex problems to global optimality using advanced algorithms (e.g., Gurobi (Gurobi Optimization, LLC, 2024), CVXPY (Diamond & Boyd, 2016)). For this reason, convexity is the widely accepted watershed between "easy" and "hard" problems. Before utilizing these solvers, the mathematical models are first represented in code as computational models, which are then passed to the solvers for optimization.

Figure 1: **Illustration of autoformulation and its challenges.** Autoformulation involves translating a problem description $d \in \mathcal{D}$ into a mathematical model $m \in \mathcal{M}$. This model is then transformed into a program that can be executed by solvers (e.g., Gurobi).

## 2.1 AUTOFORMULATION: PROBLEM DEFINITION

Boyd & Vandenberghe (2004) aptly recognized that *"the challenge, and art, in using convex optimization is in recognizing and formulating the problem. Once this formulation is done, solving the problem is ... (almost) technology"*. While solver technology has significantly matured, the process of formulating optimization models remains largely human expertise driven. Building on this insight, we define *autoformulation* as the automated process of transforming natural language descriptions of real-world problems into formal optimization models. This process aims to bridge the gap between human-readable problem statements and computational models suitable for optimization solvers, thus automating the "challenge and art" of problem formulation.

> **Formal Definition**
>
> Let $\mathcal{D}$ represent the space of natural language problem descriptions. Additionally, let $\mathcal{M}$ and $\mathcal{C}$ represent the space of all possible mathematical formulations and space of all possible computational models of optimization problems. Specifically, each $c \in \mathcal{C}$ is a computational representation (e.g. Python code), which includes choice of solving algorithm and its configurations. Given a problem $d \in \mathcal{D}$, autoformulation involves two transformations:
>
> 1. **Mathematical Formulation** $p_\phi : \mathcal{D} \rightarrow P(\mathcal{M})$: Transforming problem description into a precise mathematical formulation. Here, $P(\cdot)$ represents the space of probability distributions.
> 2. **Computational Representation** $p_\psi : \mathcal{M} \rightarrow P(\mathcal{C})$: Converting the mathematical formulation into computational formats suitable for solvers. This includes encoding the model in a programming framework and specifying or configuring an appropriate solving algorithm.
>
> **Autoformulator.** Here, $p_\phi$ and $p_\psi$ are models of each transformation, with $\phi$ and $\psi$ their respective parameters. The complete autoformulation process can thus be represented as inferring the joint distribution $p_{\phi,\psi}(m, c \mid d) = p_\psi(c \mid m) \cdot p_\phi(m \mid d)$. We refer to any algorithm designed for the autoformulation problem as an *autoformulator*.
>
> **Objective.** For a given problem $d$, autoformulation aims to find optimal mathematical and computational formulations that maximize an evaluation measure $Q(\cdot)$ :
>
> $$(m^*, c^*) \in \arg\max_{m \in \mathcal{M}, c \in \mathcal{C}} Q(m, c; d) \qquad (2)$$
>
> **Evaluation criteria.** Here, $Q$ assesses the quality of $(m, c)$ relative to $d$, considering factors such as correctness, solvability, and efficiency. The quality of a formulation is primarily evaluated based on **formulation correctness**—how accurately it reflects the problem description. Given that a formulation is correct, two additional criteria come into play. **Optimality gap:** the gap between the value of the objective function at the solution and the optimal value. For example, convex problems and certain non-convex problems like mixed-integer linear programs (MILPs) can be solved efficiently to global optimality (i.e. zero optimality gap). **Computational efficiency:** evaluating the resource requirements and solution time of a particular model. These latter two criteria are only meaningful for formulations that correctly capture the problem description.

**Challenges.** A closer scrutiny of Eq. (2) reveals a few key challenges:

**[C1] Problem-dependent hypothesis space:** Given a problem $d$, we define the *problem-dependent* hypothesis space of an autoformulator as $\mathcal{H}_{\phi,\psi}(d)$.[a] This space encompasses all plausible formulations, including correct, incorrect, or trivially redundant formulations. Manually specifying this hypothesis is highly challenging, due to complex interdependencies, the vast number of formulations, and the problem-specific nature of the space.

**[C2] Efficient search under uncertainty:** Efficiently navigating the problem-dependent hypothesis space is challenging, as performant formulations (e.g. efficient and globally optimal) can be sparse. This search involves managing two key sources of uncertainty: **a) modeling decisions:** uncertainty in the optimal way to formulate the problem, **b) problem ambiguity:** uncertainty due to ambiguous requirements such as implicit or 'common-sense' constraints (e.g. non-negativity or integer constraints for individual resources). An additional complexity is **trivial model equivalence**—formulations that are identical but have minor syntactic differences (e.g. functions $2x + 3y$ and $3y + 2x$). Note, we term these 'trivial' to distinguish this type of equivalence from mathematically equivalent reformulation such as converting a non-convex constraint to convex. Exploring these trivial variations inefficiently can lead to overlooking more diverse and potentially valuable formulations.

**[C3] Model evaluation:** While solvers can assess solvability and computational efficiency, they cannot evaluate **formulation correctness**—whether the model accurately represents all requirements in the problem description. This lack of feedback on semantic correctness significantly complicates the search process, as an efficient solution to an incorrectly formulated problem is ultimately invalid.

---

[a]For notational simplicity, we omit the subscript $(\phi, \psi)$, when the meaning is clear.

**A few observations.** Our definition of both steps as probabilistic distributions includes deterministic mappings as a special case (i.e. Dirac delta distributions). This formalism recognizes the inherent uncertainty in autoformulation, stemming from language ambiguity, domain knowledge limitations, and variability in modeling decisions. Additionally, the second transformation is assumed to be independent of the problem description $d$. In general, the first step (formulating mathematical models, $p_\phi$) presents a significantly greater challenge than the second step (creating computational models, $p_\psi$). The former requires deep domain understanding, the ability to abstract real-world complexities into mathematical constructs, and creativity in choosing effective problem formulations. In contrast, formulating computational models often follows more standardized patterns, with several commercial packages already offering automation in translating mathematical models into solver-compatible code (Fourer et al., 1990). However, the second transformation can present distinct challenges that introduce uncertainty, most notably through problem-specific choices and configurations for solvers.

**Types of optimization problems.** Additionally, we note that the challenges faced by an autoformulator fundamentally depends on the nature of the problem $d$, particularly its their convexity properties and reformulation possibilities. While some problems are naturally convex and enable direct solution for global optimality, others begin as non-convex and require the autoformulator to either identify equivalent convex reformulations or develop appropriate relaxation strategies that balance optimality with computational efficiency. In the interest of completeness, we provide a detailed categorization of different challenges presented by different problem types in App. E

## 3 LLM-ENHANCED MCTS FRAMEWORK FOR AUTOFORMULATION

Building upon our analysis of key challenges in autoformulation, we present a novel approach that takes initial steps towards addressing them. In the scope of this work, our method primarily focuses on evaluating **formulation correctness**—ensuring that autoformulated models accurately represent the problem description. Moreover, we mainly focus on the first transformation, as we observed that the second transformation was relatively straightforward for problems in available benchmarks. Instead, we implement the second transformation using a custom deterministic parser (represented as `parser`, and detailed in App. A), which successfully parsed all available problems. By concentrating on these fundamental aspects, we aim to catalyze the development of more advanced and computationally efficient autoformulators that address more challenging problems of each type.

**Overview.** At a high level, our method integrates *Large Language Models* (LLMs) within a *Monte-Carlo Tree Search* (MCTS) framework, founded on two key insights: **1)** leveraging the inherent hierarchical structure of optimization modeling by decomposing the autoformulation process into distinct stages, enabling systematic exploration using a tailored MCTS algorithm, and **2)** employing LLMs in two specialized roles: as dynamic hypothesis generators in each formulation stage, and as approximate evaluators of formulation correctness. This approach harnesses LLMs' extensive domain knowledge and contextual understanding to implicitly create and search a problem-dependent hypothesis space. This strategy bypasses the need for manual specification of the search space, which would be intractable due to the vast number of possible formulations. Furthermore, it utilizes LLMs' reasoning capability to evaluate formulation correctness against problem descriptions, providing an approximate yet meaningful signal to guide the search process. **Note:** In the interest of space, we present detailed information about all prompts used in the algorithm in App. A, providing only high-level details in the following subsections.

## 3.1 STRUCTURED DECOMPOSITION OF AUTOFORMULATION

Optimization modeling is inherently complex, involving multiple interconnected components. To manage this complexity and improve search efficiency, we propose a decomposition of the formulation process. This approach allows us to sequentially explore each model component rather than searching for entire formulations at once, potentially leading to more efficient search.

Specifically, we structurally decompose the autoformulation process into four distinct stages, each represented by $m_i$. The complete mathematical formulation is defined as $m = \oplus_i^4 m_i$, where $\oplus$ denotes the composition of model components: $m_1$—**parameters and decision variables**, $m_2$—**objective function**, $m_3$—**equality constraints**, and $m_4$—**inequality constraints**. Given a problem description $d$, the joint distribution $p_{\phi,\psi}(c, m \mid d)$ is decomposed hierarchically:

$$p_{\phi,\psi}(c, m \mid d) = p_\psi(c \mid m) \prod_{i=1}^4 p_\phi(m_i \mid m_{<i}, d) \tag{3}$$

Here, $p_\phi(m_i \mid m_{<i}, d)$ represents the sequential nature of mathematical formulation, where each component $m_i$ depends on the *partial formulation* $m_{<i} = \oplus_{j=0}^{i-1} m_j$ (with $m_0 = \emptyset$) and the problem description $d$. Additionally, the term $p_\psi(c \mid m)$ represents the computational model's dependency on the completed mathematical formulation.

## 3.2 MCTS-BASED AUTOFORMULATOR

Having established a structured decomposition of the autoformulation process, we now address the challenge of efficiently navigating this hierarchical space. We employ an MCTS-based algorithm, which is particularly well-suited for exploring complex, hierarchical search spaces (Coulom, 2006). Our MCTS constructs a search tree of depth $4$ to explore possible formulations, where each of the four levels corresponds to a component in our structured decomposition ($m_1$ to $m_4$). Nodes in this tree contain component formulations, and a complete formulation is represented by a path from the root to a terminal node, with each path yielding a unique formulation.

The MCTS algorithm iteratively builds the search tree through four key steps: ▶ **selection**, ▶ **expansion**, ▶ **evaluation**, and ▶ **backpropagation**. For notational clarity, we denote a tree node as $n$ and any of its child nodes as $n_{child} \in Child(n)$, where $Child(n)$ is the set of all child nodes of $n$. We use $\vec{n}$ to represent the *partial* formulation contained in the path from root to node $n$. For instance, $\vec{n}$ for a node of depth $2$ is the partial formulation containing the parameters, decision variables, and the objective function. Terminal nodes are denoted as $n_t$.

### 3.2.1 SELECTION

The selection step guides the search towards promising regions of the tree. Starting from the root, the algorithm recursively selects child nodes using the Upper Confidence Bound for Trees (UCT):
$n^*_{child} = \arg\max_{n_{child} \in Child(n)} \left( V(n_{child}) + \omega \sqrt{\frac{\ln N(n)}{N(n_{child})}} \right)$ (Kocsis & Szepesvári, 2006). This process continues until reaching an unexpanded node. Here, $n^*_{child}$ is the selected child node, $V(n_{child})$ is its estimated value, $N(n)$ and $N(n_{child})$ are visit counts for the parent and child nodes respectively and $\omega$ is an exploration constant. This formula balances exploitation (first term, favoring high-value nodes) with exploration (second term, favoring less-visited nodes).

### 3.2.2 EXPANSION

Upon reaching an unexpanded node $n$, we generate its child nodes $Child(n)$ through an expansion process. Unlike traditional MCTS, which typically operates within a predefined space, our expansion step explores an *undefined* hypothesis space of component formulations. To address this challenge, we employ LLMs as *dynamic hypothesis generators*, conditioned on the partial formulation constructed up to this level, to propose potential formulations for the next component. Our process involves two key steps: first, generating a diverse set of candidate formulations, and then pruning this set to remove trivial equivalences, thereby ensuring a manageable and meaningful search space.

**Context-aware hypothesis generator.** At node $n$, the LLM generates potential child nodes (next component formulations) by conditioning on the partial formulation and problem description: $p_\phi(n_{child} \mid \vec{n}, d)$. When expanding nodes at depth $i$, this is equivalent to $p_\phi(m_i \mid m_{<i}, d)$. We represent formulations using JSON format, where keys are descriptive labels and values are mathematical expressions. For example, when generating possible inequality constraints, the LLM might return the formulation: {"material_balance": $x_1 + x_2 \leq 100$, "quality_requirement": $0.8x_1 + 0.6x_2 \geq 75$}. The LLM is queried through a structured prompt with three elements: **(1) Problem description:** the original natural language problem description $d$; **(2) Partial formulation:** the current partial formulation $m_{<i}$ in JSON format; **(3) Level-specific instructions:** guidelines for the current modeling stage, including output format and relevant considerations. Note that we also request the LLM return potential formulations using the same dictionary format. For each node expansion, we sample $H \in \mathbb{N}$ hypotheses from the LLM's distribution: $\widehat{Child}(n) = \{\tilde{n}_{child}^{(h)} \mid \tilde{n}_{child}^{(h)} \sim p_\phi(\cdot \mid \vec{n}, d), \forall h \in [H]\}$, where $\tilde{n}_{child}^{(h)}$ represents the $h$-th candidate component formulation.

**Approximate pruning.** To ensure diversity in our search space and avoid redundant explorations, we prune candidates containing **trivially equivalent formulations**: $Child(n) = $ pruning($\widehat{Child}(n)$). This process removes formulations with minor syntactic differences that could lead to inefficient searches, while preserving meaningful reformulations. For this purpose, we employ *Satisfiability Modulo Theories* (SMT) solvers to check pairwise equivalence of formulations (Barrett & Tinelli, 2018). SMT solvers determine the satisfiability of logical formulas with respect to background theories. We represent objective functions and constraints as systems of equations or inequalities (where objective functions form a single-equation system). For two such systems $S_1$ and $S_2$ over variables $x$, we check the satisfiability of $\neg(\forall x\ (S_1(x) \iff S_2(x)))$, where $\neg$ denotes negation. Unsatisfiability of this formula proves equivalence, as it indicates there exists no $x$ such that the systems differ. Conversely, satisfiability indicates the systems are distinct. We apply this check to each pair of candidate formulations in $\widehat{Child}(n)$, pruning those deemed trivially equivalent. This approach balances maintaining a diverse search space with computational efficiency. We detail the exact formulae used for SMT equivalence checks in App. A.

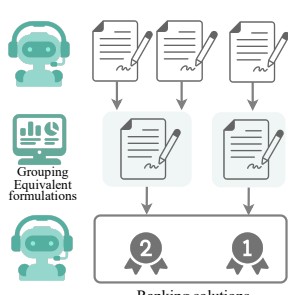

Grouping Equivalent formulations

Ranking solutions

Figure 2: **Expansion and node evaluation.** Expansion involves generating candidate formulations, which are then pruned based on functionality, and remaining formulations are assigned a normalized ranking score.

We note that the satisfiability problem in SMT is not universally decidable (Monniaux, 2016). While linear arithmetic over real and integer domains is generally decidable, mixed-integer domains or non-linear functions may be undecidable, depending on specific problem properties. When the solver cannot reach a conclusion, we assume the formulations are distinct. This approach trades efficiency for thoroughness, potentially exploring some equivalent formulations while avoiding premature pruning of unique components. Importantly, SMT solvers are only applicable to systems defined over the same variable domains. Consequently, we employ them solely for pruning redundant nodes in levels $m_2$–$m_4$, where child nodes share decision variables. For level $m_1$, which involves defining different variable domains, we utilize LLMs as approximate checkers.

### 3.2.3 EVALUATION

After expanding a node, each newly created child node undergoes an initial evaluation to estimate its value, guiding subsequent selection. This evaluation is non-trivial, as assessing the correctness of a partial formulation with respect to the original problem description is complex. Our method employs an LLM-based ranking evaluation for each set of child nodes to provide more informed initial evaluations. Specifically, we rank the partial formulation from root to each child node, namely $\{\vec{n}_{child} \mid n_{child} \in Child(n)\}$. These ranks are then center-normalized to $[0, 1]$, with the middle rank centered at $0.5$. We denote this normalized score $s(\vec{n}_{child})$, which is used to initialize the child node's value $V_{prior}(n_{child}) \leftarrow s(\vec{n}_{child})$. We note that this approach diverges from traditional MCTS, which often uses *uniform priors* for expanded nodes. Here, the LLM assesses the formulation based on optimization principles and the specific problem context, potentially capturing aspects such as formulation correctness, constraint feasibility, and alignment with the problem.

### 3.2.4 BACKPROPAGATION

**Reward.** Unlike conventional MCTS, which typically simulates the problem to a terminal state after expanding a child node, our approach continues expanding until a terminal node $n_t$ is reached, where $\vec{n}_t$ represents a complete formulation. This is computationally feasible due to our tree's limited depth of 4 levels, and provides more accurate rewards. We evaluate the complete formulation to obtain a *reward* $r(\vec{n}_t)$. We evaluate the complete formulation using a dual approach, combining assessments of both mathematical correctness and computational model's performance:

$$r(\vec{n}_t) = \mathbb{I}\left(E_{\texttt{solver}}^c(\texttt{parser}(\vec{n}_t)) = 1\right) \cdot E_{\texttt{LLM}}^m(\vec{n}_t; d) \tag{4}$$

where $\mathbb{I}$ is the indicator function. $E_{\texttt{LLM}}^m(\vec{n}_t; d)$ is the LLM's evaluation of the mathematical formulation's correctness, assessing how well it captures the problem requirements and constraints in $d$. $E_{\texttt{solver}}^c(\texttt{parser}(\vec{n}_t))$ is the solver's evaluation of the computational model's performance, providing a binary signal of whether the model is solved optimally. Note that $E_{\texttt{solver}}^c(\texttt{parser}(\vec{n}_t))$ is an imperfect signal, as an incorrectly formulated model could be solved to optimality despite not faithfully representing the original problem, highlighting the importance of our dual evaluation approach. The computational model $c$ is derived using our custom deterministic parser: $c = \texttt{parser}(\vec{n}_t)$.

Here, it is inappropriate to use the same evaluation measure (based on ranking) as described before, as the reward score would need to be comparable across all terminal nodes (across different subtrees). As such, we introduce a comparative evaluation method to obtain an LLM-evaluated correctness score. This approach compares each formulation with a baseline, asking the LLM for its preference. Specifically, the LLM returns a score $\in [0, 1]$, where $< 0.5$ values indicate preference for the baseline formulation, and value greater than $0.5$ favor the formulation in $\vec{n}_t$. Mathematically, we represent this as $E_{\texttt{LLM}}^m(\vec{n}_t; d) \sim p_{\texttt{LLM}}(\cdot \mid \vec{n}_t, m_b; d)$, where $m_b$ is the baseline formulation for comparison. This baseline serves as a consistent reference point for all child nodes, enabling more stationary and comparable evaluations of relative formulation correctness. In our implementation, we generate the baseline formulation $m_b$ through zero-shot prompting of the LLM.

**Backpropagation.** Following the reward calculation, we backpropagate this value to update the statistics of all nodes along the trajectory. For each node in this path from root to terminal node $n_t$, we apply the following updates: $V_{back}(n) \leftarrow \frac{V_{back}(n) \cdot N(n) + r(\vec{n}_t)}{N(n)+1}$, $\quad N(n) \leftarrow N(n)+1$, $\quad \forall n \in \vec{n}_t$. Here, we increment the visit count $N(n)$ by 1 and update the value $V(n)$ with a weighted average of its previous value and the new reward $r(\vec{n}_t)$. This backpropagation process ensures that the tree gradually accumulates more accurate estimates of node values. These updated statistics then inform the selection strategy in subsequent iterations. The value of the node used for selection is then $V(n) = \lambda \cdot V_{prior}(n) + (1 - \lambda) \cdot V_{back}(n)$.

**Summary.** Our MCTS-based algorithm iterates through the aforementioned steps, progressively constructing and refining a tree of possible formulations. We execute this process for $T \in \mathbb{N}$ iterations, thoroughly exploring the space of potential models and identifying promising formulations. The final output is a set of $M \in \mathbb{N}$ *functionally distinct* optimization models, where $M \leq T$. Each model is defined by a unique trajectory through the tree. Formally, we express the overall algorithm as: $\{(m^{(i)}, c^{(i)}, r^{(i)})\}_{i=1}^M = \texttt{MCTS}_{\texttt{LLM}}(d)$. The superscript $i$ indexes the functionally distinct formulation, and $r^{(i)}$ is the estimated value/reward of the corresponding terminal node.

## 4 RELATED WORK

**Advances in LLMs.** Recent works have demonstrated the substantial potential of Large Language Models (LLMs) in solving complex reasoning tasks, including language understanding (Hendrycks et al., 2021), commonsense reasoning (Brown, 2020), logical reasoning (Wei et al., 2022; Yao et al., 2024), mathematical problem-solving (Lewkowycz et al., 2022), and coding tasks (Chen et al., 2021). Of particular relevance are studies employing LLMs in optimization and search tasks, such as Bayesian Optimization (Liu et al., 2024b), prompt optimization (Guo et al., 2023), evolutionary optimization (Yang et al., 2024; Liu et al., 2024a), and symbolic program refinement (Madaan et al., 2024), as well as research exploring the integration of LLMs with planning algorithms (Huang et al., 2022; Zhao et al., 2024; Hao et al., 2023a; Zhou et al., 2024).

**Autoformulation.** Ramamonjison et al. (2023) introduced an early competition focused on translating natural language descriptions of linear programming problems into mathematical formulations. The competition involved two tasks: tagging problem entities and predicting formulations in predefined formats. Entries primarily used pre-LLM NLP models tailored for these tasks, which lacked the ability to generalize beyond the given formats. More recently, Xiao et al. (2023) and AhmadiTeshnizi et al. (2024) explored multi-agent LLM frameworks for optimization model formulation, where LLM agents generated complete formulations in each iteration, refining them locally in subsequent steps. Our approach differs by breaking down the formulation process into key stages and using MCTS to systematically explore the formulation space. Additionally, we guide the search with a composite reward function that combines solver feedback with LLM evaluation.

## 5 EXPERIMENTS

The experiments aim to evaluate the performance of the autoformulation model across a diverse set of problems (Sec. 5.1). In addition, we study two key factors of our framework: (a) the use of LLM-based evaluation to ensure the correctness of formulations (Sec. 5.2), and (b) enhanced efficiency by reducing search space in equivalent formulations (Sec. 5.3). Finally, the experiments examine failure modes to better understand the limitations of the model (Sec. 5.4).

We use NL4OPT (Ramamonjison et al., 2023) and IndustryOR as benchmarks to evaluate our approach. NL4OPT is a standard dataset for operations research tasks, primarily focusing on linear programming problems. We used the filtered version presented in (Tang et al., 2024), consisting of 244 problems. IndustryOR, with 100 problems covers linear, integer, mixed-integer, and non-linear programming across three difficulty levels, offers a more complex challenge. This complexity is crucial for testing the robustness of our method in navigating larger, ambiguous solution spaces. Since neither benchmark provides enough information to fully evaluate optimization models—lacking ground truth for decision variables, objectives, and constraints—we adopt an approximately correct framework, measuring accuracy using a ground truth value found in (Tang et al., 2024), following the same methodology. All experiments were performed using GPT4o-mini, except for NL4OPT, where GPT4-1106 was used for fair comparison.

### 5.1 BENCHMARK COMPARISON

For a comprehensive evaluation, we cover all relevant previous approaches, including: Reflexion (Shinn et al., 2023), Chain-of-Experts (Xiao et al., 2023), and OptiMUS (AhmadiTeshnizi et al., 2024) in Table 1. These approaches employ agents to iteratively enhance both the mathematical formulation and the optimization outcomes, delivering strong results. For consistency, we report the performance of these methods based on both GPT-4 outputs. We also include methods based on fine-tuning through large synthetic data such as ORLM (Tang et al., 2024). Note that our approach is not directly comparable to

Table 1: Benchmark comparison

| Method | NL4OPT | IndustryOR |
|---|---|---|
| *Finetuned methods* | | |
| ORLM-LLaMA-3-8B | 85.7% | 38.0% |
| *Methods based on GPT-4* | | |
| Standard | 47.3% | 28.0% |
| Reflexion | 53.0% | - |
| Chain-of-Experts | 64.2% | - |
| OptiMUS | 78.8% | - |
| MCTS (iter-1) | 85.24% | 35.0% |
| MCTS (iter-3) | 92.21% | 42.0% |
| MCTS (All) | 92.62% | 48.0% |

previous approaches since it captures all possible interpretations of problems. We report the accuracy of the first solution found by MCTS, as well as the accuracy of the first three solutions and after 16 rollouts. At each step, we limit the number of generated solutions to 10, with a maximum of three children per node. A higher computational budget could potentially yield more solutions.

## 5.2 EVALUATING CRITIC CAPABILITIES OF LLM

We evaluate the critic capabilities of the LLM, which in our method is used to search the formulation space. We study backpropagation (global score) and prior score (local score) independently. **(1) Getting a global score.** The score in Eq. (4) compares complete formulations. Unlike prior scoring, we compare full formulations without sharing partial structures. To isolate this component, we extract all solutions from IndustryOR that contain correct answers and compare them to wrong answers. Our goal is to show that correct solutions are generally preferred over incorrect ones when comparing formulations directly. Using this approach, we obtain a point biserial correlation coefficient of $0.48$ with a significant p-value of $2.0681 \times 10^{-3}$. We also compare this ranking method with a scoring method from the literature Zhang et al. (2024b), where formulations are scored from 1 to 100, yielding a correlation of $0.2325$ with a p-value of $1.1185 \times 10^{-1}$, which is not significant. **(2) Getting a Local Scores**.

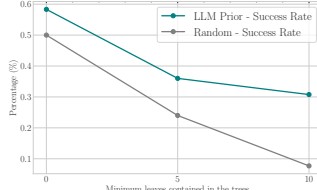

To measure prior scores, we use a simple greedy evaluation. To isolate the effect of the MCTS framework, we first construct a tree using Depth-First Search (DFS) with up to three children per node, and assign prior rewards via the LLM. For evaluation, we only consider trees containing the ground truth and measure the success rate using a greedy approach. Figure 3 shows the comparison with random prior scoring, illustrating that the chances of reaching the ground truth decrease as the number of tree leaves increases.

Figure 3: Prior score eval.

## 5.3 EFFICIENCY OF DETECTING FUNCTIONALLY EQUIVALENT FUNCTIONS

We evaluate the efficiency of our framework by analyzing the number of generated formulations at each step and the remaining formulations after two key filtering stages: (1) grouping equivalent formulations using SMT to eliminate redundancy, and (2) selecting the top three solutions based on rankings provided by the LLM. The top of Fig. 4 shows the initial number of generated formulations as a percentage (ten per experiment in practice) and tracks the results of each filtering step across all problems in IndustryOR.

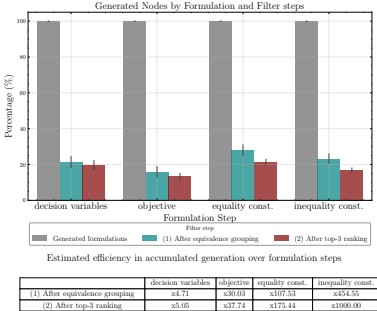

The equivalent grouping step is the most impactful, reducing the solutions by a factor of five accross all steps. The final filtering step, which selects the top-ranked formulations, discards very few solutions—a positive outcome as we aim to retain viable formulations. Based on these results, the bottom of the figure contrasts efficiency with a non-hierarchical method, where errors in earlier stages compound and affect later steps. The results demonstrate that by the time the formulations reach the inequality constraint step, efficiency increases a thousandfold. In other words, a non-hierarchical approach without filtering would have required a thousand more simulations to arrive at the same number of solutions.

Figure 4: **(Top)** Number of nodes filtered. **(Bottom)** Estimated efficiency.

## 5.4 FAILURE MODES

We examine the failure modes of our method by (1) analyzing its search capabilities for detecting the ground truth and (2) identifying the problems where the model itself fails. For the first point, Fig. 5 illustrates the search evolution of our MCTS relative to the number of rollouts, highlighting our method's ability to benefit from additional exploration (more iterations) to discover more correct solutions. Despite the excellent capabilities of our MCTS in finding novel solution, we notice that as our MCTS is sensible to recommendation of best candidates based on greedy approaches as shown in the Figure 8 of Appendix. This indicates the better critic methods are necessary to improve recommendation of best candidates.

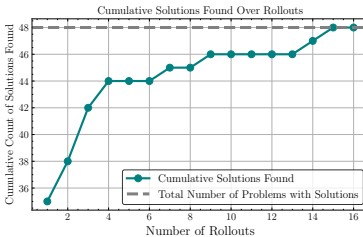

Figure 5: MCTS rollouts

We analyze failure modes based on the difficulty and problem characteristics in IndustryOR. Accuracy is measured by whether the mode finds the correct solution. In both grouping methods, the model with the highest accuracy also produces less entropic (less dense) trees. Similarly, when grouped by difficulty, the medium category exhibits lower accuracy and higher tree entropy. Our method does not show a significant weakness against any specific type of problems. For example, ORLM performs well in IPs but is weaker in MIPs (see App. D).

Table 2: Analysis by categories

| Category | Accuracy | Entropy |
|---|---|---|
| *Grouped by Difficulty* | | |
| Easy | 0.6750 | 1.9638 |
| Medium | 0.2895 | 3.0408 |
| Hard | 0.5000 | 2.7337 |
| *Grouped by Type* | | |
| IP | 0.5484 | 1.6536 |
| LP | 0.4167 | 2.1716 |
| MIP | 0.5161 | 3.3201 |

## 6 DISCUSSIONS

In this paper, we formally define the problem of autoformulation for mathematical optimization models, establishing objectives, evaluation metrics, and a categorization of problem types based on their challenges to autoformulators. We introduce a novel approach that frames autoformulation as a search problem and effectively leveraging the hierarchical structure of optimization modeling. Our method integrates LLMs as hypothesis generators and evaluation functions of formulation correctness within an MCTS framework, systematically exploring the vast hypothesis space of possible formulations. The introduction of hypothesis pruning using SMT solvers further enhances efficiency by eliminating redundant formulations. Empirical evaluations across linear and mixed-integer programming benchmarks demonstrate our method's superior performance in formulating correct models, with notable efficiency gains from pruning and LLM-based correctness evaluation.

**Future works.** Looking ahead, we envision autoformulation as an exciting domain where LLMs can significantly augment human experts. Future research directions include developing collaborative frameworks to synergize with human expertise, exploring advanced LLM-based methods such as retrieval-augmented generation (Lewis et al., 2020) and test-time compute (Lightman et al., 2024). A particularly promising direction lies in fine-tuning models to enhance autoformulator capabilities, with special emphasis on process-supervised learning for hierarchical modeling steps (Lightman et al., 2023; Wan et al., 2024), which aligns naturally with the structured decomposition inherent in optimization formulation. An important consideration for future work is the potential correlation bias from using the same LLM for both generation and evaluation; while our composite evaluation strategy helps mitigate this through solver feedback and comparative ranking, developing specialized evaluation models or ensemble-based approaches could provide more robust assessment. To support these advancements, the development of large-scale, diverse benchmarks encompassing various problem types and complexities, particularly those requiring intricate reformulations, will be crucial.

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

# A  ADDITIONAL DETAILS ON METHOD

## A.1  FORMULATION EQUIVALENCE CHECKS

SMT solvers offer a powerful approach for verifying equivalence between various components of optimization models (Barrett & Tinelli, 2018). These tools can rigorously check if different formulations of objective functions, sets of equality constraints, or sets of inequality constraints are logically equivalent. By encoding the components as logical formulas within appropriate theories (such as linear arithmetic), SMT solvers can determine if the formulations are satisfiable under the same conditions. For objective functions, the solver can check if the difference between two functions is always zero across the feasible region. This is formally described in Eq. (5). For constraint sets, it can verify if they define identical feasible regions by checking that each constraint in one set is implied by the other set and vice versa, formally described in Eqs. (6) and (7). This approach not only ensures the correctness of model transformations or reformulations but also aids in identifying redundant constraints and simplifying complex models. However, the effectiveness of SMT solvers in this context depends on the nature of the optimization problem, as nonlinear or highly complex formulations may pose challenges for current solvers.

1. For objective functions $f^{(i)}$ and $f^{(j)}$:

$$\text{Equivalent}(f^{(i)}, f^{(j)}) \iff \forall \mathbf{x} \in \mathcal{X}, f^{(i)}(\mathbf{x}) = f^{(j)}(\mathbf{x}) \tag{5}$$

2. For sets of equality constraints $g^{(i)} = \{g_k^{(i)}\}_k^K$ and $g^{(j)} = \{g_l^{(j)}\}_l^L$:

$$\text{Equivalent}(g^{(i)}, g^{(j)}) \iff \forall \mathbf{x} \in \mathcal{X}, (\bigwedge_k g_k^{(i)}(\mathbf{x}) = 0) \iff (\bigwedge_l g_l^{(j)}(\mathbf{x}) = 0) \tag{6}$$

3. For sets of inequality constraints $h^{(i)} = \{h_k^{(i)}\}_k^K$ and $h^{(j)} = \{h_l^{(j)}\}_l^L$:

$$\text{Equivalent}(h^{(i)}, h^{(j)}) \iff \forall \mathbf{x} \in \mathcal{X}, (\bigwedge_k h_k^{(i)}(\mathbf{x}) \le 0) \iff (\bigwedge_l h_l^{(j)}(\mathbf{x}) \le 0) \tag{7}$$

## A.2  PROMPT DESIGN

**Template instruction**

```
I have a problem in operational research:
------
###PROBLEM DESCRIPTION###
------
I have the following formalization:
formalization_dict = {"parameters": {}, "decision_variables2: {},
"objective": {}, "equality_constraints": {}, "inequality_constraints":
{}}
```

**Parameters template (root node generation)**

```
You are an optimization modeling expert.  Complete
formalization_dict based on the problem description, you should
complete the "parameters" field, which consists of assigning
constants to descriptive variable names.
Only complete "parameters" and nothing else.  Follow these
guidelines:

1.  Your primary responsibility is to define all the parameters
from the problem description that will later be used to define
decision variables, the objective, and constraints (both
equality and inequality).
```

```
2.  You may include additional parameters in a format suitable
for facilitating the subsequent tasks of defining decision
variables, the objective function, and constraints.
3.  For parameters that involve multiple indices (e.g., x[i] or
x[i,j]), use the most appropriate data structure, such as lists,
dictionaries, or dictionaries with tuple keys, to represent
them.
4.  For each parameter, include a clear, descriptive comment
explaining its meaning.
5.  Ensure that the parameter names (keys) are descriptive and
intuitive.

Return only the python dictionary update (i.e.,
formalization_dict["parameters"] = ...  )  following the
described requirements.
```

**Decision Variables Template (Depth == 1)**

```
You are an optimization modeling expert.  Complete only the
"decision_variables" field within the "formalization_dict" based
on the provided problem description.
Ensure the decision_variables comprehensively cover all
essential elements to accurately model the optimization problem.

Each key-value pair in the dictionary must adhere to the
following structure:

<key>: {
    "description": <description>,
    "type": <type>,
    "iteration_space": <space>
}

The structure should meet these requirements:

1.  Each <key> represents a decision variable that will later
be used to implement the objective, equality, and inequality
constraints in a Python program.
2.  Replace <key> with a symbolic name representing the decision
variable.  Ensure that each <key> represents a distinct decision
variable with a unique symbolic name.
3.  Replace <description> with a detailed explanation of the
role of the decision variable in the optimization model.
4.  Replace <type> with a string representing the Gurobi
variable type (e.g., GRB.INTEGER), as this will be used to
create the variable via Gurobi's addVar function.
5.  If the decision variable is indexed, replace <space> with a
string representing Python for-loop using list comprehension
syntax to represent the index space.  For this, assume
direct access to these parameter variables (i.e., avoid using
parameters[variables] syntax).
6.  If the variable is not indexed, set <space> to None.
7.  If the variable is indexed, do not write the index in the
symbol (do not put the index when writing <key>).
8.  You are encouraged to create decision variables that are
general.  If two decision variables represent the same concept
write them as one key, creating an appropriate iteration space.
```

```
Return only the Python dictionary update (i.e.,
formalization_dict["decision_variables"] = ...)  following the
described requirements.
```

**Objective Template (Depth == 2)**

```
You are an optimization modeling expert.  Complete only the
"objective" field within the "formalization_dict" based on the
provided problem description.
Do not complete any other fields.  Follow these requirements:

1.  Write the objective function mathematically using decision
variables.
2.  Preface the key-value pair with a Python comment explaining
the rationale behind the objective.  DO NOT make a commentary
inside the mathematical description.
3.  Use parameter-defined variables instead of hard-coded
values.  Assume direct access to these parameter variables
(i.e., avoid using parameters[variables] syntax).
4.  The dictionary key must be 'min' or 'max', reflecting the
nature of the objective (minimization or maximization).
5.  The dictionary value must be a string representation of the
objective function based on the problem description, written in
valid Python syntax.

Return only the Python dictionary update (i.e.,
formalization_dict["objective"] = "max":  ...  or
formalization_dict["objective"] = "min":  ...)  following the
described requirements.
```

**Equality Constraints Template (Depth == 3)**

```
You are an optimization modeling expert.  Complete the
formalization_dict by filling in the equality_constraints field
based on the problem description and the decision variables
provided.
These constraints include border constraints, initialization,
and equality constraints derived from the problem description.
Do not complete the "inequality_constraints" field.  Follow
these requirements:

1.  Descriptive constraints:  Each key in the dictionary should
represent a unique, clearly named constraint, with the value
being a string that describes the corresponding mathematical
equality using "==".
2.  Parameter Variables:  Use parameter-defined variables
instead of hard-coded values.  Assume direct access to these
parameter variables (i.e., avoid using parameters[variables]
syntax).
3.  Indexed Variables:  For indexed decision variables, indicate
the index within brackets (e.g., x[i]).
4.  Handling Multiple Constraints:  For similar constraints that
repeat across indices or variables, use Python for loops and
list comprehensions for efficient representation.
5.  String mathematical description:  Note, the value
(mathematical description) should be a single string.  DO NOT
use .join() or anything else.  Even if it represents multiple
constraints using a for loop.
```

6. No Inequality Constraints: Only define equality
constraints. Inequality constraints will be handled separately
by a subsequent expert.
7. Comments: Include a Python comment before each key-value
pair, explaining the rationale behind the constraint.

Return only the Python dictionary update (i.e.,
formalization_dict["equality_constraints"] = ...) following
these requirements.
Important: If the problem contains only inequality
constraints and no equality constraints, return:
formalization_dict["equality_constraints"] = {None: None}.
This will signal the need to focus on inequality constraints
in subsequent modeling steps.

**Inequality Constraints Template (Depth == 4)**

You are an optimization modeling expert. Complete the
formalization_dict by adding the inequality_constraints field
based on the problem description. Follow these requirements:

1. Descriptive constraints: Each key in the dictionary should
represent a unique, clearly named constraint, with the value
being a string that describes the corresponding mathematical
inequality.
2. Parameter Variables: Use parameter-defined variables
instead of hard-coded values. Assume direct access to these
parameter variables (i.e., avoid using parameters[variables]
syntax).
3. Indexed Variables: For indexed decision variables, indicate
the index within brackets (e.g., x[i]).
4. Handling Multiple Constraints: For similar constraints that
repeat across indices or variables, use Python for loops and
list comprehensions for efficient representation.
5. String mathematical description: Note, the value
(mathematical description) should be a single string without
using join or anything else. Even if it represents multiple
constraints using a for loop.
6. Inequality Constraints Only: Include only inequality
constraints. Exclude any constraints already covered under
equality_constraints.
7. Comments: Include a Python comment before each key-value
pair, explaining the rationale behind the constraint.

Return only the Python dictionary update (i.e.,
formalization_dict["inequality_constraints"] = ...) following
these requirements.
Important: Think carefully of inequality constraints that
are not explicit in the problem description that should be
considered. If after thinking you conclude the problem contains
only equality constraints and no inequality constraints, return:
formalization_dict["inequality_constraints"] = {None: None}.

**Group Decision Variables template**

- Objective:

```
As an expert in optimization modeling, your role is to evaluate
multiple sets of decision variables provided for an operations
research problem.  You are responsible for determining if two or
more sets of decision variables should be grouped together based
on their equivalency from an optimization perspective.
- Task Breakdown:

Your grouping decision is critical for assisting a subsequent
optimization expert, who will define the objective function,
equality constraints, and inequality constraints for each group.
To facilitate this process, follow these precise guidelines:

- Equivalency Criteria:

1.  Same Objectives and Constraints:  Two sets of decision
variables should be grouped together if they result in the
definition of the same objective function, equality constraints,
and inequality constraints, even if the variable names differ.
2.  Conceptual Equivalency:  Variable sets should be grouped
together if, despite having different variable names, they
define the same underlying concepts that ultimately lead
to identical objectives and constraints (both equality and
inequality).
3.  Non-Equivalency Conditions:  Two sets of decision variables
should not be grouped together if they lead to differences in
any of the following:  Objective function, Equality constraints,
Inequality constraints.
4.  Naming Convention Irrelevance:  The names of the decision
variables are irrelevant for grouping purposes.  Only the
functional impact of the variables on the objective function
and constraints should be considered.  If two sets of variables
lead to the same results, group them together, even if the names
differ.

By following these guidelines, you will help ensure that
decision variable sets are clearly classified for the next
expert in the process.

Please list your clusters as follows:

###
groups = {
1: group_1,
...,
n: group_n}
###

Where group_i is a python list containing the names (string) of
all the set of decision variables that are equivalent.  One set
of decision variables can only belong to one group.  The list
should consider at least one element.

Important:  Think carefully STEP BY STEP about your grouping
decision, then conclude your assessment using the structured
format provided above.

Here are the current solutions:

solutions = {}
```

**Ranking Template to obtain prior rewards**

```
You are an expert in optimization modeling.  Using the
formalization_dict as your current progress, you are tasked with
selecting the optimal #VARIABLE# from the provided options.

Please follow these steps:

1.  Carefully evaluate each potential #VARIABLE#.

2.  Rank the variables from best to worst based on their
suitability.

Present your rankings in the following format:

###
rank = {
1: solution_1,
...,
n: solution_n}
###

Where:
- solution_1 represents the best #VARIABLE#.
- solution_n represents the least suitable #VARIABLE#.

Important:  Think carefully STEP BY STEP about your ranking
decision.  Then conclude by listing the solutions in string
format as structured above.

Here are the possible solutions:

solutions = {}
```

### A.3 EVALUATION METRICS

#### EXPECTED CORRECTNESS

**Mathematical Definition:** Expected Correctness (EC) quantifies the probability of reaching a correct leaf node when starting from a given node and following a probabilistic policy based on the node rankings $R$. It is computed recursively:

- For a leaf node $n$:

$$\text{EC}(n) = \begin{cases} 1, & \text{if } n \text{ is correct} \\ 0, & \text{if } n \text{ is incorrect} \end{cases}$$

- For an intermediate node $n$ with children $\{c_i\}$:

$$\text{EC}(n) = \sum_i p(c_i \mid n) \times \text{EC}(c_i)$$

where the probability of choosing child $c_i$ is given by:

$$p(c_i \mid n) = \frac{R(c_i)}{\sum_j R(c_j)}$$

**Intuitive Explanation:** Expected Correctness measures the likelihood that, by following the ranking-based probabilities at each decision point, we will eventually arrive at a correct solution. It reflects the overall effectiveness of the ranking system in guiding the search process toward correct outcomes.

PRECISION AND RECALL

**Mathematical Definition:**

- **Precision** at a node is the ratio of relevant children (those leading to correct leaves) to all retrieved children, averaged over all positions:

$$\text{Precision} = \frac{1}{n} \sum_{k=1}^{n} \frac{\text{Relevant}_k}{k}$$

where $\text{Relevant}_k$ is the number of relevant children in the top $k$ positions.

- **Recall** at a node is the ratio of relevant children retrieved to the total number of relevant children, averaged over all positions:

$$\text{Recall} = \frac{1}{n} \sum_{k=1}^{n} \frac{\text{Relevant}_k}{\text{Total Relevant}}$$

**Intuitive Explanation:** Precision indicates how well the top-ranked children (based on $R$) correspond to those that lead to correct solutions. High precision means most top-ranked choices are relevant. Recall measures the ability of the ranking to capture all relevant children among its selections. High recall implies that the ranking method successfully identifies most of the correct paths.

NORMALIZED DISCOUNTED CUMULATIVE GAIN (NDCG@K)

**Mathematical Definition:** NDCG@k evaluates the quality of the ranking up to position $k$, accounting for the position of relevant items:

1. Compute DCG@k (Discounted Cumulative Gain):

$$\text{DCG@}k = \sum_{i=1}^{k} \frac{2^{\text{rel}_i} - 1}{\log_2(i+1)}$$

where $\text{rel}_i$ is the relevance score at position $i$ (1 if the child leads to a correct leaf, 0 otherwise).

2. Compute IDCG@k (Ideal DCG) by ordering the children perfectly:

$$\text{IDCG@}k = \sum_{i=1}^{k} \frac{2^{\text{rel}_i^*} - 1}{\log_2(i+1)}$$

where $\text{rel}_i^*$ is the ideal ordering of relevance scores.

3. NDCG@k is the ratio:

$$\text{NDCG@}k = \frac{\text{DCG@}k}{\text{IDCG@}k}$$

**Intuitive Explanation:** NDCG@k assesses not just whether relevant children are present in the top $k$ positions but also how highly they are ranked. It rewards rankings that place relevant children earlier, reflecting the practical importance of quickly finding correct solutions.

TOP-RANK SUCCESS RATE

**Mathematical Definition:** The Top-Rank Success Rate is the probability that always choosing the top-ranked child (highest $R$) at each decision point leads to a correct leaf:

$$\text{Top-Rank Success Rate} = \frac{\text{Number of times top-path leads to correct leaf}}{\text{Total number of root nodes}}$$

**Intuitive Explanation:** This metric evaluates the effectiveness of a greedy strategy based solely on the ranking $R$. A high success rate indicates that the top-ranked paths are reliable guides to correct solutions, simplifying decision-making in the tree traversal.

## B  COMPARISON TO RELATED WORKS: MCTS METHODS

Table 3: **Comparison of LLM+MCTS variants.** This table presents a comparative analysis of LLM+MCTS methods, highlighting the methodological innovation of our method. Specifically, we compare **(1) expansion mechanisms** for generating child nodes; sources of evaluation for **(2) newly expanded nodes**; and **(3) terminal nodes**; whether **(4) evaluations employ comparative evaluations**; and finally the **(5) search space**. $\mathcal{V}, \mathcal{E}, \mathcal{N}$ represent evaluation signals obtained from LLMs, external environments (e.g. test cases in coding tasks), and comparisons with other solutions, respectively. $\mathcal{T}, \mathcal{P}, \Sigma$ represent space of natural language thoughts, formal programs, and tokens, respectively.

| Method | Expansion mechanism | Initial node evaluation | Reward used in backpropagation | Evaluation via comparisons? | Search space |
|---|---|---|---|---|---|
| RAP (Hao et al., 2023b) | - | $\mathcal{V}$ | $\mathcal{V}$ | ✗ | $\mathcal{T}$ |
| LATS (Zhou et al., 2023) | - | $\mathcal{V}$ | $\mathcal{V}, \mathcal{E}$ | ✗ | $\mathcal{T}$ |
| MCTSr (Zhang et al., 2024b) | Self-refine | $\mathcal{V}$ | $\mathcal{V}$ | ✗ | $\mathcal{T}$ |
| VeriGenMCTS (DeLorenzo et al., 2024) | Filter by functionality | - | $\mathcal{E}$ | ✗ | $\Sigma$ |
| RethinkMCTS (Li et al., 2024) | Rethink | $\mathcal{V}, \mathcal{E}$ | $\mathcal{V}, \mathcal{E}$ | ✗ | $\mathcal{T}$ |
| VerMCTS (Brandfonbrener et al., 2024) | Joint Expansion-Evaluation | $\mathcal{E}$ | $\mathcal{E}$ | ✗ | $\mathcal{P}$ |
| AlphaZero like MCTS (Feng et al., 2023b; Zhang et al., 2024a; Chen et al., 2024a; Luo et al., 2024) | - | $\mathcal{V}$ | $\mathcal{V}, \mathcal{E}$ | ✗ | $\mathcal{T}$ |
| DPO like MCTS (Xie et al., 2024; Chen et al., 2024b) | - | $\mathcal{V}$ | $\mathcal{V}, \mathcal{E}$ | ✗ | $\mathcal{T}$ |
| **Ours** | SMT Pruning | $\mathcal{V}, \mathcal{E}, \mathcal{N}$ | $\mathcal{V}, \mathcal{E}, \mathcal{N}$ | ✓ | $\mathcal{M}_{1:4}$ |

Recent advancements in MCTS have demonstrated that combining MCTS with LLMs significantly enhances reasoning capabilities by refining the thinking process (Zhang et al., 2024b). Complementary improvements have been achieved using Process Reward Models, which further optimize reasoning using training with step-wise reward information (Feng et al., 2023a). In parallel, the integration of MCTS and LLMs has been applied to code generation (Zhong et al., 2023; Brandfonbrener et al., 2024; Li et al., 2024). We present a detailed comparison against these related works in Tab. 3.

Our MCTS framework introduces three key innovations specifically tailored to autoformulation:

1. **Structured hierarchical search:** We leverage the inherent structure of optimization modeling to decompose the search space. Unlike conventional MCTS approaches, which assume fixed search spaces, our hierarchical organization of search spaces both reduces search complexity and increases formulation diversity.
2. **SMT-based pruning:** Our analysis shows that 80% of generated formulations are trivially equivalent (see Fig. 4). By integrating SMT solvers to prune these redundant formulations, we achieve a 400x improvement in search efficiency, avoiding exponential growth in search complexity.
3. **Comparative formulation evaluation:** We introduce novel pairwise comparative evaluation for assessing formulation correctness, which is distinct from the standard approach where LLMs evaluate solutions in isolation. This comparative framework enables more reliable preference-based evaluation to improves search efficiency.

## C  REFORMULATION STRATEGIES

Here, we enumerate commonly used strategies to reformulate the problem. Detailed descriptions can be found in (Boyd & Vandenberghe, 2004, Chapter 4.1.3).

- Change of variables
- Transformation of functions
- Slack variables
- Eliminating equality constraints
- Adding equality constraints
- Optimizing over some variables
- Epigraph form

# D  ADDITIONAL RESULTS

## D.1  OBTAINING LOCAL SCORES (PRIOR RANKING).

We evaluated our node ranking method on the IndustryOR dataset, aggregating metrics across all problems. The assessment used five key metrics: Average Expected Correctness, Average Precision, Average Recall, NDCG@3, and Top-Rank Success Rate. Average Expected Correctness measures the likelihood of reaching a correct leaf by following a probabilistic policy based on RR scores. Precision and Recall assess the accuracy and coverage of ranking correct subtrees. NDCG@3 evaluates ranking quality within the top three positions, and Top-Rank Success Rate gauges the success of a greedy strategy that always selects the top-ranked child. Table in Figure 6 presents detailed results, with the top section showing Precision and Recall by formulation step and the bottom displaying aggregated results across all nodes.

Our results shows an Average Expected Correctness of 0.6692 indicating a significant likelihood of reaching correct solutions using the probabilistic policy based on $\mathbb{R}$ scores. The high Average Precision (0.8911) and Recall (0.8643) show that our ranking method effectively prioritizes correct subtrees while identifying most of them. The NDCG@3 of 0.6776 reflects strong ranking performance within the top three positions, crucial when resources for exploration are limited. Finally, the Top-Rank Success Rate that 73% of the time selecting the top-ranked child leads to correct solutions. These results validate the robustness of our ranking approach in guiding the search process toward correct solutions.

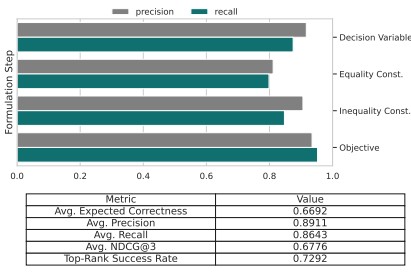

| Metric | Value |
|---|---|
| Avg. Expected Correctness | 0.6692 |
| Avg. Precision | 0.8911 |
| Avg. Recall | 0.8643 |
| Avg. NDCG@3 | 0.6776 |
| Top-Rank Success Rate | 0.7292 |

Figure 6

## D.2  ASSESSING THE CRITIC EVALUATION CAPABILITIES OF LLMS

**Prior rewards.** To evaluate whether the prior reward serves as a reliable indicator of a good node during node selection, we compute the *normalized accumulated prior rewards*. This is the sum of all prior rewards from the visited nodes, normalized by dividing the total by 4 (corresponding to 4 steps). Using this metric, we analyze the results across the IndustryOR dataset, grouping the outcomes based on whether the formalization produces a score equal to the ground truth or not. The results of this study is presented in Figure 7. The mean accumulated reward for successful formalizations (*prediction equals ground truth*) is 0.744 ($\sigma_{succ} = 0.133$), notably higher than the mean for failed formalizations, which is 0.651 ($\sigma_{fail} = 0.122$). An independent t-test confirms that this difference is statistically

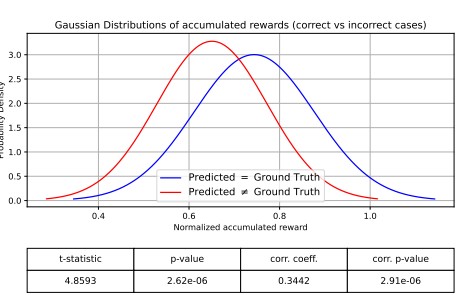

| t-statistic | p-value | corr. coeff. | corr. p-value |
|---|---|---|---|
| 4.8593 | 2.62e-06 | 0.3442 | 2.91e-06 |

Figure 7: **Top**. Distribution of *accumulated prior rewards* of correct prediction vs failed prediction. **Bottom** Statistical test comparing both groups.

significant ($t = 4.86, p < 0.001$), indicating that successful formalizations consistently yield higher rewards. Additionally, a point-biserial correlation analysis shows a moderate positive correlation ($r = 0.344, p < 0.001$) between accumulated rewards and success, suggesting that higher accumulated rewards are associated with an increased likelihood of arriving at the successful formalization.

## D.3 ADDITIONAL COMPARISONS AGAINST ORLM

Table 4: Comparison between GPT4-omini. For the MCTS approach the accuracy if the ground truth is found by the MCTS search.

| Method | Difficulty | | | Question Types | | | | |
|---|---|---|---|---|---|---|---|---|
| | Easy | Medium | Hard | LP | NLP | IP | MIP | Others |
| ORLM-LLaMA-3-8B | 57.5% | 20.0% | 35.0% | 36.1% | 0.0% | 61.2% | 19.3% | 0.0% |
| MCTS (ours) | 67.50% | 28.95% | 50.00% | 41.67 % | 0.0% | 54.84% | 51.61% | 0.0% |

## D.4 GREEDY SEARCH AFTER 16 ROLLOUTS MCTS

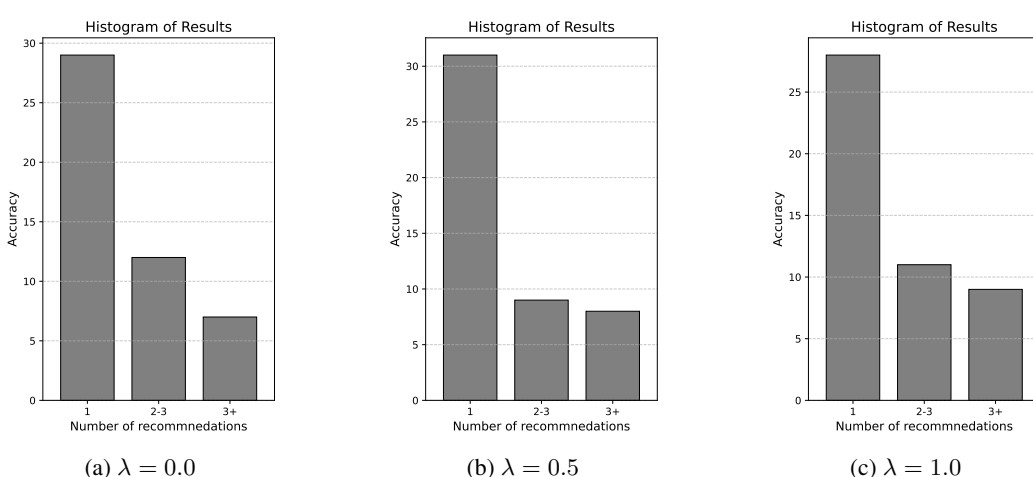

(a) $\lambda = 0.0$      (b) $\lambda = 0.5$      (c) $\lambda = 1.0$

Figure 8: Initially, we perform MCTS with 16 rollouts. Following this, we apply a greedy search to determine the number of iterations required to find the ground truth solution.

## D.5 ADDITIONAL ANALYSIS ON PARTIAL FORMULATION EVALUATIONS

| Formalization Step | Correlation | p-value |
|---|---|---|
| Decision Variables | 0.1985 | 0.1159 |
| Objective | 0.2143 | 0.2470 |
| Equality Constraints | 0.3711 | 0.0004 |
| Inequality Constraints | 0.3290 | 0.0033 |

Table 5: Spearman's Correlation and p-values for different formalization steps.

**Experimental setup.** To analyze partial model evaluation, we compute correlations between intermediate node evaluations in the MCTS and ground-truth correctness scores. For each intermediate node, we define its ground-truth correctness as the percentage of correct leaf nodes in its subtree.

**Observations.** Analysis of Spearman's correlations (Tab. 5) reveals two key patterns:

1. **Evaluation robustness increases with depth.** Correlation strength improves from decision variables ($r = 0.1985$) to inequality constraints ($r = 0.3290$), with corresponding gains in statistical significance. This aligns with intuition: deeper nodes contain more complete formulation information, enabling more accurate evaluation.

2. **Component-specific evaluation challenges.** Earlier components show weaker correlations due to interdependencies between modeling elements. For instance, evaluating decision variables in isolation is challenging without understanding their role in objectives and constraints. This increased uncertainty at earlier stages reflects the inherent difficulty in assessing partial formulations without full context.

These findings underscore the importance of hierarchical search strategies that maintain diverse exploration paths, particularly in early stages where evaluation signals are weaker.

Table 6: Impact of **formulation** and **solver** on optimality gap and computational efficiency.

| | Factor | Optimality gap | Computational efficiency |
|---|---|---|---|
| **Type I (originally convex)** | Formulation | **Minimal impact**: any correct (equivalent) formulation can achieve global optimality | **Medium impact**: choice of variable representation (e.g. structure-preserving formulations) can improve solution time |
| | Solver | **Minimal impact**: most commercial solvers can achieve similar optimality gaps | **Medium impact**: specialized solvers for specific problem structures (LP, QP, SOCP) can be faster |
| **Type II (non-convex, but convexifiable)** | Formulation | **High impact**: correct, convexified reformulation enables achievable global optimality | **High impact**: reformulation complexity affects solution time, generally convex reformulations are solved faster |
| | Solver | **Medium impact**: solver ability to handle reformulated structures affect solution quality | **High impact**: solver must efficiently handle the specific structure of reformulation |
| **Type III (non-convex, requiring relaxation)** | Formulation | **High impact**: quality of relaxation directly affects optimality gap and bounds tightness | **Medium impact**: relaxation complexity affects solution time, involving trade-offs between relaxation tightness and computational efficiency |
| | Solver | **High impact**: solver abilities on relaxed problem affects solution quality | **Medium impact**: solving relaxed problems may require specialized solvers, although general purpose solvers are roughly comparable |

# E    CATEGORIZATION OF AUTOFORMULATION CHALLENGES BY OPTIMIZATION PROBLEM STRUCTURE

The exact challenges faced by an autoformulator depends on the nature of the problem $d$. Here, we provide a categorization of optimization problems and their characteristics. To help elucidate different types of problems, we introduce two concepts. First, we define the **set of correct formulations** for a problem $\mathcal{M}(d) \subset \mathcal{M}$ as the set of all equivalent formulations that *correctly* model a problem $d$. Second, we introduce the set of **original forms** $\mathcal{M}_o(d) \subseteq \mathcal{M}(d)$—the set containing the natural representations of the problem, typically the initial models an optimization expert would create. This is a set, as it can contain trivially equivalent formulations. Finally, we partition the set $\mathcal{M}$ into the set of convex problems $\mathcal{M}_{\text{conv}}$ and the set of non-convex problems $\mathcal{M}_{\text{nonc}}$.

1. **Type I problems.** These are problems where the original form is inherently convex, namely $\mathcal{M}_o(d) \subseteq \mathcal{M}_{\text{conv}}$. Examples include certain resource allocation problems that can be naturally formulated as linear programs. The challenge of solving Type I problems is to ensure that the problem is correctly represented (**formulation correctness**, i.e. $\mathcal{H}(d) \cap \mathcal{M}_o(d) \neq \emptyset$), which would entail that it can be efficiently solved to global optimality.

2. **Type II problems.** These are problems where the original form is non-convex, but can be reformulated into an equivalent convex problem, namely $\mathcal{M}_o(d) \subseteq \mathcal{M}_{\text{nonc}}$ but $\mathcal{M}(d) \cap \mathcal{M}_{\text{conv}} \neq \emptyset$. In addition to formulation correctness, another challenge of solving Type II problems is to ensure the autoformulator can identify and apply appropriate reformulation strategies (e.g. change of variables) to transform the non-convex into an *equivalent* convex form, namely $\mathcal{H}(d) \cap (\mathcal{M}(d) \cap \mathcal{M}_{\text{conv}}) \neq \emptyset$ (see App. C for discussion). For such problems, evaluation extends beyond correctness to include the ability to achieve **global optimality** through reformulation.

3. **Type III problems.** These are problems where the original form is non-convex and cannot be reformulated into a convex problem, namely $\mathcal{M}(d) \subseteq \mathcal{M}_{\text{nonc}}$. In such cases, there are two general options: a) solve the non-convex problem using general-purpose algorithms (e.g. gradient descent), or b) *relax* into a convex problem that approximates, but is not equivalent to, the original problem (e.g. semidefinite relaxation of a Max-Cut problem (Goemans & Williamson, 1995)).

A crucial nuance here is that mathematically equivalent models, even when both are convex, can exhibit vastly different computational complexities. An example of this is quadratic programming and second-order cone programming (SOCP) reformulations of the same problems (Alizadeh & Goldfarb, 2003). Although mathematically equivalent, SOCP formulations often allow for more efficient solution methods. Therefore, **computational efficiency** is an important evaluation metric across all three problem types, significantly impacting practical utility of model formulations. In App. F, we provide concrete examples to illustrate each type of optimization problems.

# F    ILLUSTRATIVE EXAMPLES OF PROBLEM CATEGORIZATION

In this section, we provide examples of canonical problems in engineering and machine learing that belong to each of the identified problem types. Specifically:

- **Type I**: Problems that have a precise mathematical model, which is convex in its original form. Examples are provided in App. F.1.
- **Type II**: Problems that have a precise mathematical model, which is nonconvex in its original form but can be reformulated as a convex problem (sometimes additional assumptions are needed). Examples are provided in App. F.2.
- **Type III**: Problems that have a precise mathematical model, which is nonconvex in its original form but can be *relaxed* to a convex problem (sometimes additional assumptions are needed). The difference from **Type II** is that the convex relaxation is *not* equivalent to the original problem. Examples are provided in App. F.3.

We also provide examples of problems where the problem description is inherently ambiguous in App. F.4

## F.1    EXAMPLES OF **TYPE-I** PROBLEMS

**[P1] Mutual Information Maximization** (`Mutual-Information`)

- **Reformulation strategies:** None.
- **Difficulty in reformulation:** Easy.
- **Difficulty in solving the reformulated problem:** Easy.

Mutual information is a quantity that measures the divergence between two random variables, with applications in wireless communications (Goldsmith & Varaiya, 1996) and in data science (Belghazi et al., 2018). Here we describe it in the context of maximizing Shannon capacity in wireless communications.

We consider a discrete memoryless channel with an input random variable $X \in \{1, \ldots, \ell\}$, an output random variable $Y \in \{1, \ldots, y\}$, and channel transition matrix $P \in \mathbb{R}^{y \times \ell}$ with the element on the $j$-th row and the $i$-th column being $p_{ji} = \mathrm{prob}\,(Y = j \mid X = i)$.

$$\text{Input } X \longrightarrow \boxed{\text{Transition Probability } P} \longrightarrow \text{Output } Y$$

Our goal is to choose the optimal probability distribution of input $X$, denoted $x \in \mathbb{R}^\ell$ with $x_i = \mathrm{prob}\,(X = i)$, in order to maximize the mutual information between input $X$ and input $Y$

$$I(X;Y) = \sum_{i=1}^{\ell} \sum_{j=1}^{y} x_i p_{ji} \log_2 \frac{p_{ji}}{\sum_{k=1}^{\ell} x_k p_{jk}}.$$

The optimal value of the problem is called Shannon capacity.

This problem is convex in its original form:

$$
\begin{aligned}
\text{Maximize} \quad & \sum_{i=1}^{\ell} \left( \sum_{j=1}^{y} p_{ji} \log_2 p_{ji} \right) x_i - \sum_{j=1}^{y} \left( \sum_{i=1}^{\ell} p_{ji} x_i \right) \log_2 \left( \sum_{i=1}^{\ell} p_{ji} x_i \right) \\
\text{subject to} \quad & x_i \geq 0, \quad i = 1, \ldots, \ell, \\
& \sum_{i=1}^{\ell} x_i = 1.
\end{aligned}
\tag{8}
$$

**[P2] Power control for maximum throughput** (`PC-MaxRate-decoupled`)

- **Reformulation strategies:** None.

- **Difficulty in reformulation:** Easy.
- **Difficulty in solving the reformulated problem:** Easy.

We consider the problem of allocating a unit amount of total power over $\ell$ independent frequency or temporal wireless channels (Yu & Lui, 2006; Luo & Zhang, 2008). Each channel has a channel gain $g_i$ and a noise power $\sigma_i$, resulting in a channel gain to noise ratio of $\alpha_i = g_i / \sigma_i$.

Our goal is to choose the optimal power allocation, denoted $x \in \mathbb{R}_+^\ell$ with $x_i$ being the power allocated to channel $i$, in order to maximize the total throughput.

This problem is convex in its original form:

$$
\begin{aligned}
\text{Maximize} \quad & \sum_{i=1}^{\ell} \log_2 \left( 1 + \alpha_i x_i \right) \\
\text{subject to} \quad & \sum_{i=1}^{\ell} x_i \leq 1, \\
& x_i \geq 0, \quad i = 1, \dots, n.
\end{aligned}
\tag{9}
$$

**[P3] Power control to satisfy SINR requirements with minimum power** (`PC-MinPower`)

- **Reformulation strategies:** Transformation of function.
- **Difficulty in reformulation:** Easy (straightforward observation).
- **Difficulty in solving the reformulated problem:** Easy (the reformulated problem is LP).

We consider the problem of determine the transmit power of $\ell$ pairs of transceivers. They operate in the same frequency at the same time, hence causing interference to each other. The problem data is a channel gain matrix $\mathbf{G} \in \mathbb{R}^{\ell \times \ell}$, where $g_{ij}$ is the channel gain from transmitter $j$ to receiver $i$, the noise power vector $\sigma \in \mathbb{R}^\ell$ with $\sigma_i$ as the noise power at receiver $i$, and the minimum SINR requirement vector $\gamma \in \mathbb{R}^\ell$ with $\gamma_i$ as the minimum SINR required by transceiver $i$.

Our goal is to choose the transmit power, denoted $x \in \mathbb{R}_+^\ell$ with $x_i$ being the power of transmitter $i$, in order to minimize the total transmit power while satisfying the SINR requirements of each transceiver (Yates, 1995).

This problem is non-convex in its original form:

$$
\begin{aligned}
\text{Minimize} \quad & \sum_{i=1}^{\ell} x_i \\
\text{subject to} \quad & \frac{g_{ii} x_i}{\sum_{j \neq i} g_{ij} x_j + \sigma_i} \geq \gamma_i, \quad i = 1, \dots, \ell.
\end{aligned}
\tag{10}
$$

But it is not hard to observe that the constraints of SINR requirements can be reformulated as linear constraints, resulting in a LP:

$$
\begin{aligned}
\text{Minimize} \quad & \sum_{i=1}^{\ell} x_i \\
\text{subject to} \quad & g_{ii} x_i \geq \gamma_i \left( \sum_{j \neq i} g_{ij} x_j + \sigma_i \right), \quad i = 1, \dots, \ell.
\end{aligned}
\tag{11}
$$

**[P4] Beamforming to minimize sidelobes** (`Beamform-MinSidelobe`)

- **Reformulation strategies:** Transformation of functions, slack variables.
- **Difficulty in reformulation:** Easy (standard techniques were used).
- **Difficulty in solving the reformulated problem:** Medium (the reformulated convex problem is a second-order cone program (SOCP)).

We consider the problem of an antenna array with $\ell$ elements in a 2-D space, with the position of $k$th element denoted by the coordinate $(x_k, y_k)$. Given the complex-valued weight $w_k$ (i.e., a voltage or current phasor) of each element, the output of the array at direction $\theta$ is

$$G(\mathbf{w}; \theta) = \sum_{k=1}^{\ell} w_k e^{\mathbf{i}(x_k \cos \theta + y_k \sin \theta)},$$

where $\mathbf{i} = \sqrt{-1}$ and $G(\mathbf{w}; \theta)$ is complex-valued.

Our goal is to achieve certain gain at the target direction $\theta_{\text{tar}}$, while minimizing the energy radiated at directions $\theta_1, \ldots, \theta_m$ outside the target area (Lebret & Boyd, 1997).

This problem is convex:
$$
\begin{aligned}
\text{Minimize} \quad & \max_{i=1,\ldots,m} |G(\mathbf{w}; \theta_i)| \\
\text{subject to} \quad & G(\mathbf{w}; \theta^{\text{tar}}) = 1.
\end{aligned}
\tag{12}
$$

It is convex because the objective function is the point-wise maximum of convex functions (i.e., norms) and the constraint is linear.

However, the maximum operator makes the objective function non-differentiable. To make it amenable for standard solvers, we would reformulate it as the following SOCP:
$$
\begin{aligned}
\text{Minimize} \quad & t \\
\text{subject to} \quad & |G(\mathbf{w}; \theta_i)| \leq t, \quad i = 1, \ldots, m, \\
& G(\mathbf{w}; \theta^{\text{tar}}) = 1.
\end{aligned}
\tag{13}
$$

**[P5] Optimal power flow in direct current** (`DC-OPF`)

- **Reformulation strategies:** None.
- **Difficulty in reformulation:** Easy.
- **Difficulty in solving the reformulated problem:** Medium (the number of variables is in the order of hundreds to ten thousands).

We consider the problem of dispatching $\ell$ electric power generators in a power system. Here we use the direct current (DC) problem formulation. The decision variables are the power generation of each generator $P_i$ and the bus angle $\theta_i$.

Our goal is to minimize the total generation cost, subject to the balance of supply and demand, the power flow equations, and the power generation limits (Bakirtzis & Biskas, 2003).

This problem is convex: (Bakirtzis & Biskas, 2003)
$$
\begin{aligned}
\text{Minimize} \quad & \max_{i=1,\ldots,\ell} C_i(P_i) \\
\text{subject to} \quad & \left| \frac{1}{x_{ij}} (\theta_i - \theta_j) \right| \leq F_{ij}^{\max}, \quad \forall i, j = 1, \ldots, \ell, \\
& P_i^{\min} \leq P_i \leq P_i^{\max}, \quad i = 1, \ldots, \ell, \\
& \mathbf{B} \cdot \theta = \mathbf{P} - \mathbf{D},
\end{aligned}
\tag{14}
$$

where $C_i(\cdot)$ is the generation cost function (usually quadratic), $x_{ij}$ and $F_{ij}^{\max}$ are the admittance and the power flow limit of the transmission line connecting bus $i$ and bus $j$, $P_i^{\min}$ and $P_i^{\max}$ are the lower and upper limits of power generator $i$, $\mathbf{B}$ is the admittance matrix, and $\mathbf{D}$ is the demand vector.

## F.2 EXAMPLES OF **TYPE-II** PROBLEMS

**[P1] PC for maximum throughput with interference** (`PC-MaxRate-interference`)

- **Reformulation strategies:** Change of variables.
- **Difficulty in reformulation:** Hard (highly skilled techniques were used).
- **Difficulty in solving the reformulated problem:** Medium (the reformulated convex problem may not be recognized by standard solvers as convex).

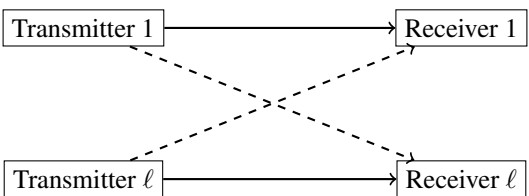

We consider the problem of determine the transmit power of $\ell$ pairs of transceivers. They operate in the same frequency at the same time, hence causing interference to each other. The problem data is a channel gain matrix $\mathbf{G} \in \mathbb{R}^{\ell \times \ell}$, where $g_{ij}$ is the channel gain from transmitter $j$ to receiver $i$, and the noise power vector $\sigma \in \mathbb{R}^\ell$ with $\sigma_i$ as the noise power at receiver $i$.

Our goal is to choose the transmit power, denoted $x \in \mathbb{R}_+^\ell$ with $x_i$ being the power of transmitter $i$, in order to maximize the total throughput.

This problem is nonconvex and is proved to be NP-hard in its original form: (Luo & Zhang, 2008)

$$
\begin{aligned}
\text{Maximize} \quad & \sum_{i=1}^{\ell} \log_2 \left( 1 + \frac{g_{ii} x_i}{\sum_{j \neq i} g_{ij} x_j + \sigma_i} \right) \\
\text{subject to} \quad & \sum_{i=1}^{\ell} x_i \leq 1, \\
& x_i \geq 0, \quad i = 1, \ldots, \ell.
\end{aligned}
\tag{15}
$$

Under a "weak interference" assumption, we can reformulate the problem as a convex problem. Specifically, we define $\mathbf{H} = \mathbf{G} + \sigma \cdot \mathbf{1}^T$, namely $h_{ij} = g_{ij} + \sigma_i$. We assume that $\mathbf{H}$ is invertible with $\mathbf{H}^{-1} \triangleq \mathbf{I} - \mathbf{C}$ and that $\mathbf{C}$ is a nonnegative matrix. This assumption holds true when $g_{ii} > \sum_{j \neq i} g_{ij}$ for all $i$, namely the interference is weak. Under this assumption, with the change of variables $\mathbf{y} = \mathbf{H}\mathbf{x}$, namely $y_i = \sum_{j=1}^{n} h_{ij} x_j$, the problem can be reformulated as the following convex problem:

$$
\begin{aligned}
\text{Maximize} \quad & \sum_{i=1}^{n} \log_2 \left( \frac{y_i}{c_i^T y} \right) \\
\text{subject to} \quad & \sum_{i=1}^{n} \left( y_i - c_i^T y \right) \leq 1, \\
& y_i - c_i^T y \geq 0, \; i = 1, \ldots, n.
\end{aligned}
\tag{16}
$$

We can prove that the objective function is concave by showing that the negative of its Hessian matrix is positive semidefinite. However, standard solvers such as CVXPY cannot recognize it as a concave function. Therefore, we need to define a custom objective function in standard solvers or write custom interior-point methods to solve it.

**[P2] Optimal power flow in alternating current and radial networks** (`AC-OPF-radial`)

- **Relaxation strategies:** Semidefinite relaxation (SDR).
- **Difficulty in reformulation:** Medium (SDR techniques were used).
- **Difficulty in solving the reformulated problem:** Hard (the number of variables is in the order of hundreds to ten thousands).

Please see `AC-OPF` in App. F.3. It is shown that under certain conditions (e.g., the power network is a radial network, which is often the case for distribution networks), the convex relaxation of the problem is exact (Lavaei & Low, 2011). In other words, the non-convex original problem is equivalent to a convex problem.

F.3 EXAMPLES OF **TYPE-III** PROBLEMS

**[P1] Beamforming with nonconvex problem definitions** (`Beamform-Nonconvex`)

- **Relaxation strategies:** Semidefinite relaxation (SDR).
- **Difficulty in reformulation:** Easy (standard SDR techniques were used).
- **Difficulty in solving the reformulated problem:** Medium (the relaxed convex problem is a SDP, and recovery methods are needed).

We consider the same setting as `Beamform-MinSidelobe`. But here our goal is to maximize the gain at the target direction $\theta_{\text{tar}}$, while limiting the ripple effect at directions $\theta_1, \dots, \theta_m$ outside the target area.

This problem is non-convex: ([Fuchs, 2013](#))

$$
\begin{aligned}
\text{Maximize} \quad & \left| G(\mathbf{w}; \theta^{\text{tar}}) \right| \\
\text{subject to} \quad & 1/\delta \le |G(\mathbf{w}; \theta_i)| \le \delta, \ i = 1, \dots, m.
\end{aligned}
\tag{17}
$$

It is non-convex because we maximize a convex function (i.e., norm) and have constraints on convex functions greater than or equal to a constant.

In this case, the standard semidefinite relaxation technique can be used, which "lifts" the problem to higher dimensions. Specifically, we define a rank-1 semidefinite matrix $\mathbf{W} \triangleq \mathbf{w}\mathbf{w}^H$. Then the gain at direction $\theta$ satisfies

$$
|G(\mathbf{w}; \theta)|^2 = \text{trace}\left(\mathbf{E}(\theta_i) \cdot \mathbf{W}\right),
$$

where $\mathbf{E}(\theta_i) = \mathbf{e}(\theta_i)^* \cdot \mathbf{e}(\theta_i)^T \in \mathbb{C}^{n \times n}$ with

$$
\mathbf{e}(\theta_i) = \left[ e^{\mathbf{i}(x_1 \cos\theta + y_1 \sin\theta)}, \dots, e^{\mathbf{i}(x_n \cos\theta + y_n \sin\theta)} \right]^T.
$$

With the new matrix variable $\mathbf{W}$, we have the following equivalent problem:

$$
\begin{aligned}
\text{Maximize} \quad & \text{trace}\left(\mathbf{E}(\theta_{\text{tar}}) \cdot \mathbf{W}\right) \\
\text{subject to} \quad & (1/\delta)^2 \le \text{trace}\left(\mathbf{E}(\theta_i) \cdot \mathbf{W}\right) \le \delta^2, \ i = 1, \dots, m, \\
& \mathbf{W} \succeq 0, \\
& \text{rank}(\mathbf{W}) = 1.
\end{aligned}
\tag{18}
$$

Here, the only nonconvexity comes from the rank constraint. By removing it, we get the following convex relaxation:

$$
\begin{aligned}
\text{Maximize} \quad & \text{trace}\left(\mathbf{E}(\theta_{\text{tar}}) \cdot \mathbf{W}\right) \\
\text{subject to} \quad & (1/\delta)^2 \le \text{trace}\left(\mathbf{E}(\theta_i) \cdot \mathbf{W}\right) \le \delta^2, \ i = 1, \dots, m, \\
& \mathbf{W} \succeq 0.
\end{aligned}
\tag{19}
$$

In general, we need to recover an approximate solution vector $\mathbf{w}$ from the solution matrix $\mathbf{W}$. Under certain conditions (e.g., uniform linear arrays), we can guarantee to recover the *exact* solution vector.

**[P2] Optimal power flow in alternating current** (`AC-OPF`)

- **Relaxation strategies:** Semidefinite relaxation (SDR).
- **Difficulty in reformulation:** Medium (SDR techniques were used).
- **Difficulty in solving the reformulated problem:** Hard (the number of variables is in the order of hundreds to ten thousands).

This is the alternating current (AC) version of `DC-OPF`, and is more accurate in power system modelling.

A canonical version of `AC-OPF` is as follows: ([Lavaei & Low, 2011](#))

$$
\begin{aligned}
\text{Minimize} \quad & \max_{i=1,\dots,\ell} C_i(V_i) \\
\text{subject to} \quad & \underline{v}_i \le |V_i|^2 \le \bar{v}_i, \quad i = 1, \dots, \ell i, j = 1, \dots, \ell, \\
& \underline{s}_j \le \sum_{k:(j,k)\in\mathcal{E}} y_{jk}^H V_j \left(V_j^H - V_k^H\right) \le \bar{s}_j, \ j \in \mathcal{N},
\end{aligned}
\tag{20}
$$

where $V \in \mathbb{C}^{\ell}$ is the vector of voltages at each bus, $C_i(\cdot)$ includes the generation cost and the power loss in transmission lines, the first constraints are voltage magnitude constraints to ensure stability of transmission lines, and the second constraints are power injection constraints derived from Ohm's law and Kirchhoff's laws.

This problem is non-convex because of the constraints. A common approach is to perform semidefinite relaxation and solve the convex relaxation. Please see (Lavaei & Low, 2011) for details.

### F.4 EXAMPLES OF **AMBIGUOUS** PROBLEMS

**[P1] Power control for maximum throughput** (`PC-MaxRate`)

This prompt does not provide the context of whether these transceivers cause interference to each other. If they are using orthogonal frequency-division multiplexing (OFDM), the major multiple-access protocol in the fourth-generation (4G) and fifth-generation (5G) communications, we can formulate the problem as the relatively easy **PC-MaxRate-decoupled**. If they are self-organizing as a mesh network, we can formulate the problem as the challenging **PC-MaxRate-interference**. In this case, the LLM should ask for the specific use case to determine the mathematical model.

**[P2] Beamforming** (`Beamform`)

The design specifications are not clear. The user may want to minimize the sidelobe while keeping certain gain at the target direction, resulting in **Beamform-MinSidelobe**, or may want to maximize the gain at the target direction while limiting the sidelobe, resulting in **Beamform-Nonconvex**. In this case, the LLM should ask for a clear design specification form the user.

**[P3] Optimal power flow** (`OPF`)

The problem statement is not clear about which version of the OPF problem to use. When the power system is not heavily loaded and when fast solutions are needed, we can use the DC approximation and formulate the problem as `DC-OPF`, which is a LP. If we need accurate solutions (e.g., solutions that inform us of the power loss on the transmission lines), we need to formulate it as `AC-OPF`, which is non-convex in general. However, if the underlying network is a distribution grid, which is a radial network, the resulting `AC-OPF-radial` has a convex relaxation, which has been proven to be exact (i.e., equivalent to the original non-convex problem).

### F.5 OVERVIEW

In Fig. 9, we visualize the difficulty in solving and reformulating the example problems.

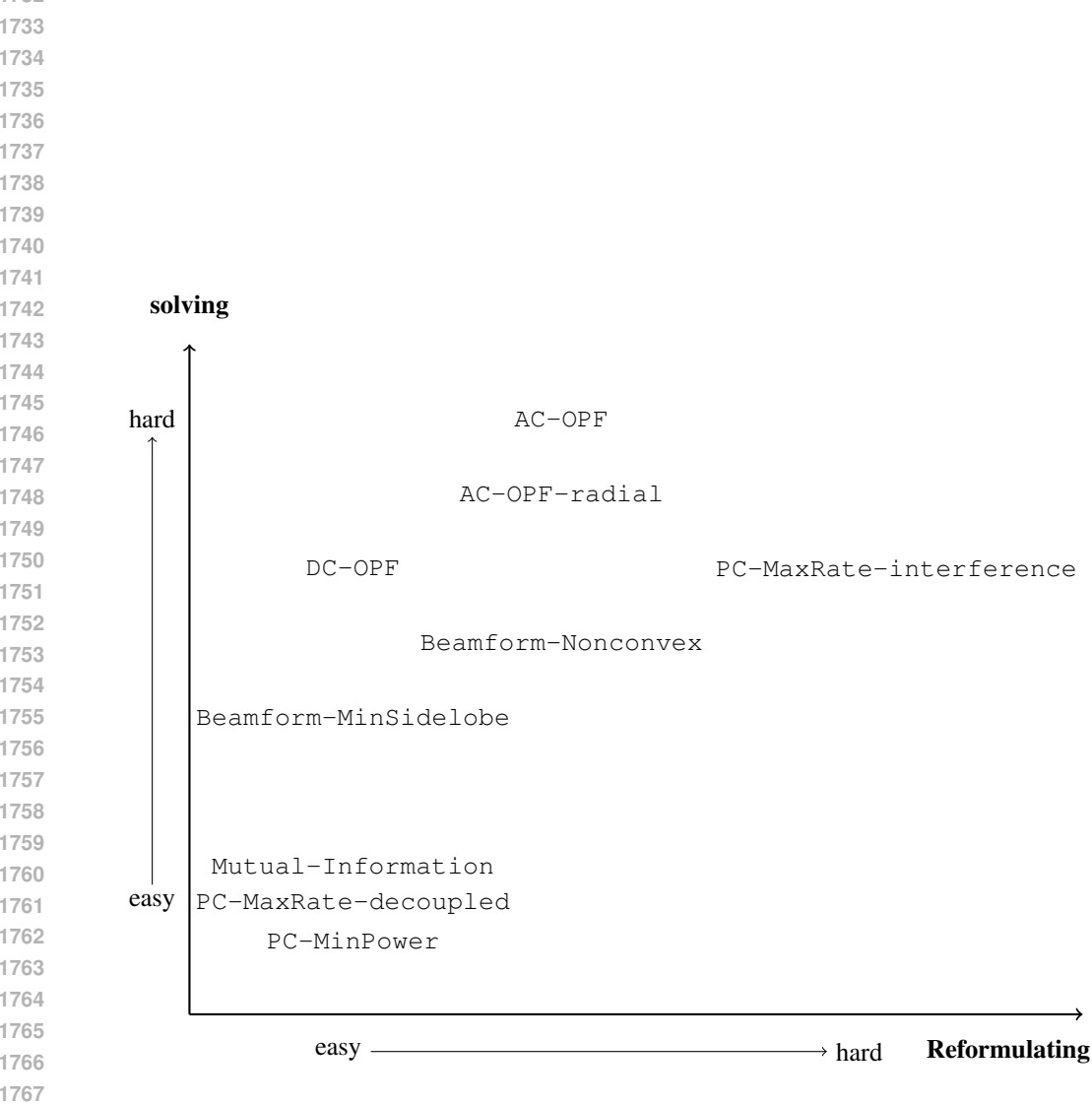

Figure 9: Example problems in terms of difficulty in reformulating and solving the problem.

# G  SOLVER PERFORMANCE COMPARISON

In this section, we present simulation results comparing the performance of various optimization solvers on the `PC_MinPower` problem, both in its original nonconvex form and in a reformulated convex form. We consider problem instances with $\ell = 10$ and $\ell = 100$ users (i.e., $\ell$ optimization variables). For each instance, we evaluate the solvers in terms of success rate, optimality gap, and average solve time over 100 random samples.

## G.1  EXPERIMENTAL SETUP

The `PC_MinPower` problem aims to minimize the total power consumption in a system while satisfying certain constraints. The original formulation of this problem is nonconvex, which can pose challenges for optimization algorithms. However, by reformulating the problem, it can be converted into an equivalent convex problem, which is generally easier to solve efficiently.

We evaluated the following solvers:

- **General-Purpose Solvers**:
  - **TRCA**: Trust-Region Constrained Algorithm
  - **SLSQP**: Sequential Least Squares Programming.
  - **COBYLA**: Constrained Optimization BY Linear Approximations.
  - **COBYQA**: Constrained Optimization BY Quadratic Approximations.
- **Convex Program Solvers**:
  - **CLARABEL**: A conic optimization solver.
  - **ECOS**: Embedded Conic Solver.
  - **SCS**: Splitting Conic Solver.
  - **OSQP**: Operator Splitting Quadratic Program Solver.

For each solver and problem instance, we recorded:

- **Success Rate**: The percentage of runs where the solver successfully found a feasible solution.
- **Optimality Gap**: The difference between the objective value obtained by the solver and the known optimal value.
- **Average Solve Time**: The average computation time (in seconds) required by the solver.

## G.2  RESULTS

Tables 7 and 8 present the performance of the solvers for problem sizes $n = 10$ and $n = 100$, respectively.

Table 7: Solver Performance for $\ell = 10$ Users

| Solver | Type | Original Nonconvex Problem | | | Reformulated Convex Problem | | |
|---|---|---|---|---|---|---|---|
| | | Success | Optimality Gap | Time (s) | Success | Optimality Gap | Time (s) |
| *General-Purpose Solvers* | | | | | | | |
| TRCA | General-Purpose | 100% | $7.31 \times 10^{-3}$ | 0.0399 | 100% | $1.25 \times 10^{-3}$ | 0.0420 |
| SLSQP | General-Purpose | 100% | $6.47 \times 10^{-7}$ | 0.0019 | 100% | $6.48 \times 10^{-7}$ | 0.0009 |
| COBYLA | General-Purpose | 67% | $2.94 \times 10^{-6}$ | 0.0073 | 100% | $7.15 \times 10^{-6}$ | 0.0039 |
| COBYQA | General-Purpose | 0% | — | — | 6% | 9.80 | 14.4067 |
| *Convex Program Solvers* | | | | | | | |
| CLARABEL | Convex Solver | — | — | — | 100% | $6.32 \times 10^{-7}$ | 0.0002 |
| ECOS | Convex Solver | — | — | — | 100% | $6.16 \times 10^{-7}$ | 0.0001 |
| SCS | Convex Solver | — | — | — | 100% | $4.45 \times 10^{-7}$ | 0.0002 |
| OSQP | Convex Solver | — | — | — | 100% | $6.48 \times 10^{-7}$ | 0.0003 |

## G.3  DISCUSSION

The results demonstrate several key observations:

Table 8: Solver Performance for $\ell = 100$ Users

| Solver | Type | Original Nonconvex Problem | | | Reformulated Convex Problem | | |
|---|---|---|---|---|---|---|---|
| | | Success | Optimality Gap | Time (s) | Success | Optimality Gap | Time (s) |
| *General-Purpose Solvers* | | | | | | | |
| TRCA | General-Purpose | 100% | $7.75 \times 10^{-2}$ | 0.6628 | 100% | $1.28 \times 10^{-2}$ | 0.6856 |
| SLSQP | General-Purpose | 100% | $1.04 \times 10^{-6}$ | 0.0750 | 100% | $1.04 \times 10^{-6}$ | 0.0298 |
| COBYLA | General-Purpose | 0% | — | 6.0764 | 100% | $2.86 \times 10^{-5}$ | 9.9629 |
| COBYQA | General-Purpose | 0% | — | — | 0% | — | — |
| *Convex Program Solvers* | | | | | | | |
| CLARABEL | Convex Solver | — | — | — | 100% | $6.22 \times 10^{-7}$ | 0.0121 |
| ECOS | Convex Solver | — | — | — | 100% | $9.91 \times 10^{-7}$ | 0.0097 |
| SCS | Convex Solver | — | — | — | 100% | $1.05 \times 10^{-6}$ | 0.0055 |
| OSQP | Convex Solver | — | — | — | 100% | $1.04 \times 10^{-6}$ | 0.0110 |

- **Importance of Problem Reformulation**: For general-purpose solvers, reformulating the original nonconvex problem into an equivalent convex problem significantly improves solution quality. This improvement is more pronounced for larger problem sizes ($\ell = 100$). For instance, COBYLA's success rate increased from 0% to 100% when the problem was reformulated.
- **Solver Selection Matters**: Different general-purpose solvers exhibit varying performance levels. SLSQP consistently achieves near-zero optimality gaps and high success rates with relatively low solve times across both problem formulations and sizes. In contrast, COBYQA fails to find feasible solutions in most cases, highlighting the necessity of careful solver selection.
- **Performance of Convex Program Solvers**: For the reformulated convex problem, convex program solvers (CLARABEL, ECOS, SCS, OSQP) show excellent and consistent performance. They all achieve 100% success rates, negligible optimality gaps, and minimal solve times. The differences among these solvers are minimal, suggesting that any of them would be suitable for solving the convex formulation efficiently.

These findings underscore the importance of problem reformulation and solver selection in optimization tasks. Reformulating a nonconvex problem into a convex one can significantly enhance the performance of general-purpose solvers. Additionally, selecting the appropriate solver is crucial, as it can greatly impact the success rate and computational efficiency.

# H    BROADER PERSPECTIVE ON AUTOFORMULATION AND OPTIMIZATION MODELING

Our autoformulation framework addresses a specific, yet crucial challenge: translating problem descriptions into mathematical and computational models. However, it is important to acknowledge that this represents just one component of the broader modeling process, which typically involves:

1. Information gathering from diverse stakeholders, often requiring multiple iterations and integration of implicit domain knowledge/conventions (i.e., 'tribal knowledge');
2. Handling complex problem characteristics such as stochasticity, time-varying dynamics, and large-scale variables;
3. Rigorous validation against real-world data, including sensitivity analysis of modeling assumptions and verification of edge cases;
4. Continuous communication between technical and business stakeholders to ensure practical utility.

Understanding these complexities helps identify which aspects of modeling can be effectively automated, thereby enabling OR practitioners to focus their expertise on more nuanced challenges such as stakeholder engagement and validation of modeling assumptions.

