# OpenReview forum: "Autoformulation of Mathematical Optimization Models Using LLMs"
_ICLR.cc/2025/Conference — Submitted to ICLR 2025_

### Official Review · Reviewer_mbTn · 2024-10-24

**Soundness:** 2
**Presentation:** 3
**Contribution:** 2
**Rating:** 5
**Confidence:** 4

**Summary:**

This paper proposed a MCTS-enhanced LLM approach, to transform descriptions of certain types of optimization problems into formal optimization models consisting of (1) variables, (2) objective, (3) equality constraints and (4) inequality constraints. The MCTS has a depth of 4 corresponding to the four components of an optimization model. When expanding a node, LLMs are used to generate a set of candidate formulations, and trivally identical formulations are pruned by SMT solvers to improve efficiency. The expanded nodes will be assigned a prior by an LLM-based ranking evaluation. The reward at the terminal node is a combination of LLM evaluation and whether optimality is reached from solvers. Experimental results show that the proposed approach outperforms other non-MCTS approaches.

**Strengths:**

1. Good writing, easy to follow.
2. Specific improvements on MCTS to fit the optimization scenario (merging trivially equivalent formulations, including solver signals to rewards, etc.)
3. Sufficient experimental details.

**Weaknesses:**

1. The rationality of the definition of "autoformulation" is not sufficiently backed by the main part of the paper. The connection between the problem setting and the proposed method is not very clear. For the definition, two transformations are highlighted (mathematical formulation model $p_\phi$ and computational representation model $p_\psi$), and "autoformulation" is defined to jointly optimize $(\phi, \psi)$ so that the expection of a quality function $Q$ is maximized. This problem setting is a bit different from other "LLM+OR" papers.  However, in the main part of the paper, no clear effort is paid on optimizing parameters, neither $\phi$ nor $\psi$ (GPT-4 is used without fine-tuning). I feel that the main objective of this paper is still the same as other papers, proposing a $P_{\phi\text{-LLM}}^\text{improved}$ that empirically performs better than a vanilla use of $P_{\phi\text{-LLM}}$. So I am a bit confused why bother to propose such a definition. Specifically, it seems unneeded to have a probabilistic model $p_\psi$ with complex joint optimization, if the computational representation step can be simply done by a deterministic parser (line 241) or commertial packages (line 194).
2. There are already lots of literatures working on combining LLMs with MCTS to solve general or domain-specific problems (see arXiv:2309.17179, arXiv:2402.03289, arXiv:2409.09584, arXiv:2406.07394, arXiv:2402.08147 for some examples), in both training and inference stage, which may unfortunately raise the bar for another "LLM+MCTS" paper to appear on top conferences. It is good to have LLM+MCTS applied on OR problems, but more unique features might be required to distinguish this paper. This paper has paid some effort on it (see Strength 2), but apart from these adaptations, the whole LLM+MCTS pipeline seems quite standard, and only applied at the inference stage.
3. While LLM contains general knowledge of optimization, the proposed method is limited to (reformulated) convex problems (type I and II in this paper).

**Questions:**

Questions:
1. Why having a probabilistic model $p_\psi$ with complex joint optimization in Section 2.1? (See Weaknesses 1 for details)

Suggestions:
1. Highlight the relation and difference between this paper and other works combining LLMs with MCTS.
2. Align the problem definition and the methods. If the quality of computational representation is not sufficiently addressed in this paper, the problem definition may not be so complex. (In my opinion, a method which fit the current problem definition will be, for example, bi-level finetuning of two LLMs $p_\phi$ and $p_\psi$ to maximize $Q$)
3. It may worth fine-tuning the model with MCTS-based self enhancement in the future (check arXiv:2309.17179 for an example).

---

> ### Author Response · Authors · 2024-11-21
> **Response to Reviewer mbTN (1/2)**
>
> *We appreciate the reviewer’s detailed and thoughtful evaluation and positive feedback.*
>
> ---
>
> ### [P1] Autoformulation objective
>
> Thank you for raising these important concerns. Allow us to clarify our original rationale, then outline our planned revisions to improve the connection between our theoretical framework and methodology.
>
> **Rationale of probabilistic definition.** The probabilistic framework (`Eq 2`) serves as a generalized formulation that:
> 1. Captures the inherent uncertainty in translating natural language into mathematical/computational models
> 2. Explicitly decouples mathematical formulation from computational implementation, recognizing their distinct challenges
>
> **Connecting definition to method.** Our method directly instantiates this objective (`Eq 2`):
> * $p_\phi$ implements a tree-search model exploring possible formulations, where $\phi$ represents the parameters that are updated through MCTS backpropagation: the value function $V(s)$ and visitation counts $N(s)$. Formally, for the set of all nodes $\mathcal{N}$, $\phi=\lbrace{(V(s), N(s)) | s \in\mathcal{N}\rbrace}$
> * $p_\psi$ is realized through our custom-designed deterministic parser, tailored to our mathematical model representation
> * $Q(\cdot)$ implements the dual evaluation of formulation correctness (based on LLM comparative scoring) and optimality gap (`Eq 4`)
>
> **Proposed revisions.** So far, we have only provided clarifications. While our probabilistic framework is theoretically sound, your review has helped us realize that the current formulation can be confusing for the reader, obfuscating the core connection between the problem definition and methodology. To better align with our method's focus on search, we propose revising `Eq 2` to:
>
> $(m^*, c^*) \in \text{argmax}_{m \in \mathcal{M}, c \in \mathcal{C}}  ~ Q(m, c; d)$
>
> This revised formulation more explicitly connects our search-based approach while maintaining generality.
>
> **Distribution over computation models.** Lastly, we wanted to clarify that the second transformation ($p_\psi$), in principle, presents several important challenges that can introduce uncertainty. Most notably, this is reflected in the choice of solvers and their configurations, for example, the configuration of hyperparameters for cutting plane algorithms. Our probabilistic formulation anticipates these challenges and provides a framework for future research to systematically address them, even though we made simplifying implementation choices (deterministic parsing) in this first exploration of autoformulation.
>
> **Actions taken:** Following your feedback, we have revised the formulation of `Eq 2` to improve clarity and expanded the discussion of computational transformation challenges in `L195`.

---

> > ### Author Response · Authors · 2024-11-21
> > **Response to Reviewer mbTN (2/2)**
> >
> > ---
> >
> > ### [P2] Comparison to related works
> >
> > We appreciate the suggestion of relevant related works. Let us clarify the distinctive aspects of our approach at multiple levels:
> >
> > **Novel problem.** Our work addresses automating mathematical optimization modeling—a significant real-world challenge where formulation errors can lead to suboptimal resource allocation, inefficient operations, and costly business decisions. One of our main contributions lies in formally defining and characterizing the unique challenges of this problem.
> >
> > **Problem-driven method design.** Our use of MCTS stems directly from the problem structure of autoformulation, not vice versa. The hierarchical nature of optimization modeling and uncertainty in optimal formulations naturally motivates MCTS as the most suitable search framework.
> >
> > **Novel methodological components.** Our MCTS framework introduces three key innovations specifically tailored for autoformulation:
> > 1. **Structured hierarchical search:** We leverage the inherent structure of optimization modeling to decompose the search space. Unlike conventional MCTS approaches, which assume fixed search spaces, our hierarchical organization of search spaces both reduces search complexity and increases formulation diversity.
> > 2. **SMT-based pruning:** Our empirical analysis revealed that $80\%$ of generated formulation hypotheses are trivially equivalent (`Fig 4`). By integrating SMT solvers to prune these redundant formulations, we achieve a 400x improvement in search efficiency, avoiding exponential growth in search complexity.
> > 3. **Comparative formulation evaluation:** We introduce novel pairwise comparative evaluation for assessing formulation correctness, which is distinct from the standard approach where LLMs evaluate solutions in isolation. This comparative framework enables more reliable preference-based evaluation to improve search efficiency.
> >
> > **Empirical impact.** Our method achieves state-of-the-art performance, finding $10\%$ more accurate formulations than ORLM (a model specifically fine-tuned for OR formulation) on the challenging IndustryOR dataset, demonstrating that our architectural innovations translate to real-world performance gains.
> >
> > **In summary.** While LLM+MCTS is indeed an active research area, our work's novelty lies in addressing a **novel and impactful problem** with distinct technical challenges through problem-driven **methodological innovations** that achieve significant **empirical improvements**.
> >
> > **Action taken:** In the interest of space, we provide an in-depth comparison with suggested related works in `App B` of the revised manuscript.
> >
> > ---
> >
> > ### Questions
> >
> > 1. **Q1:** Please see our response [P1].
> > 2. **S1:** Please see our response [P2].
> > 3. **S3:** Thank you for this interesting suggestion. Fine-tuning an autoformulator with process-reward models (PRM) is indeed promising, as it could enable data-driven improvement of hierarchical modeling while systematically capturing domain expert knowledge. We expect that the primary challenge would lie in curating a dataset with step-wise (or component-wise) rewards to train the PRM, which requires deep domain expertise for accurate labeling of intermediate modeling decisions. One potential approach to address this challenge is using outcome-based labels that can be redistributed to derive stepwise rewards [1]. We have expanded on this promising direction for future work in `L512`.
> >
> > [1] Luo, L., Liu, Y., Liu, R., Phatale, S., Lara, H., Li, Y., Shu, L., Zhu, Y., Meng, L., Sun, J. and Rastogi, A., 2024. Improve Mathematical Reasoning in Language Models by Automated Process Supervision. arXiv preprint arXiv:2406.06592.
> >
> > ---
> > *We hope the reviewer’s concerns are addressed and they will consider updating their score. We welcome further discussions.*

---

> ### Comment · Reviewer_mbTn · 2024-11-25
>
> I acknowledge that I have read the response from the authors, and appreciate their improvement on this paper. In general, I don't think this is a bad paper. The presentation is clear, the experiment is sufficient, thus I would give a high score if this is a submission to some non-top/application-oriented venues. What really concerns me is that the paper is too "standard", which is not bad for real-world engineering practice, but may be an issue for top conferences like ICLR. I try not to say something like "lack of novelty" in my review, and the authors do made some efforts as their response said, but most of the concepts and techniques are familiar in OR textbooks and other LLM papers. In my very personal opinion, the only major thing that is not so standard is that the "computational representation" stage is represented as a probabilistic model. I know that computational representation can be tricky, and different representations can impact the efficiency especially for large-scale problems, so a model that produces optimal computational representations can be a strength and really interests me. However, the paper does not address this in the method section, which is a pity.
>
> Overall, my opinion for this paper is pretty neutral. I won't be bothered at all if it gets accepted into ICLR.

---

> > ### Author Response · Authors · 2024-11-26
> > **Thank you and to summarize**
> >
> > Dear Reviewer mbTn,
> >
> > Thank you again for taking the time to review our paper and providing thoughtful and candit feedback that has improved our work. We appreciate your recognition of our paper's improved clarity and experimental rigor, while understanding that perspectives may vary.
> >
> > ---
> >
> > ### [Comment 1] On the perceived 'standard' nature of our work
> >
> > While some aspects of our methodology may appear familiar at first glance, our work makes significant novel contributions to an **emerging challenge**: *automating optimization model formulation, which remains a critical bottleneck despite decades of advances in solving algorithms*. This problem has substantial real-world impact, as enhancing formulation efficiency can lead to significant cost savings and broader access to optimization tools.
> >
> >
> > We want to emphasize that our technical contributions are:
> > 1. A formal framework for autoformulation that identifies distinct problem classes and their challenges, enabling systematic analysis of formulation automation and guiding solution development.
> > 2. A novel method introducing components specific to optimization modeling absent in related literature: **structured hierarchical search space**, **SMT-based pruning**, and **comparative formulation correctness evaluation** (as outlined in our previous response).
> > Our work moves beyond simply applying LLM+MCTS to optimization problems. Instead, it introduces a principled approach, grounded in our formal framework, that combines symbolic reasoning (through SMT solvers), structured exploration (via MCTS and hierarchical decomposition), and neural language models in ways specifically designed for mathematical optimization.
> >
> > We argue that developing novel methods, even those sharing high-level parallels with existing approaches, to solve important practical problems represents a valuable scientific contribution—particularly when such development demands innovative design choices and demonstrates measurable improvements in performance. Our state-of-the-art results on challenging benchmarks demonstrate the effectiveness of these innovations.
> >
> >
> > ### [Comment 2] On the second transformation
> >
> > We deeply appreciate your insight regarding computational representations and their impacts on solver efficiency, particularly in large-scale problems. While our paper primarily focuses on the first transformation—translating problem descriptions into mathematical formulations—this scope was chosen deliberately, as we found this to be the **most challenging frontier in automating optimization modeling**. Our empirical analysis showed that while computational representation is crucial, the primary bottleneck currently lies in correctly formulating mathematical models from problem descriptions.
> >
> > The probabilistic framing of computational transformations in our work opens interesting research directions, including automated solver selection and configuration optimization. These aspects, though not fully explored in the current paper, align with our **broader goal of providing a comprehensive framework for automated optimization modeling**, to pave the way for continued innovation in this domain.
> >
> >
> > ---
> > ### In Closing
> >
> > Thank you again for your thoughtful feedback and for recognizing our efforts. We believe our work makes meaningful contributions both **methodologically**—through novel components for LLM+MCTS integration—and **theoretically**—by providing a foundational framework for automated optimization modeling. We hope these clarifications have helped illustrate the depth and significance of these advances. We remain excited about the research directions this work enables.
> >
> >
> > Sincerely,
> >
> > The Authors of #10599

---

### Official Review · Reviewer_Utyj · 2024-10-27

**Soundness:** 2
**Presentation:** 2
**Contribution:** 2
**Rating:** 6
**Confidence:** 3

**Summary:**

This paper identifies three core challenges: defining the vast, problem-dependent hypothesis space, efficiently searching this space under uncertainty, and evaluating the correctness of formulations.

To tackle these issues, the authors propose a novel method that leverages Large Language Models (LLMs) within a Monte-Carlo Tree Search framework. LLMs function as both hypothesis generators and evaluators of formulation correctness, while a pruning technique is employed to eliminate trivial equivalent formulations.

Empirical evaluations demonstrate that this approach significantly improves the efficiency and accuracy of formulating optimization models for linear and mixed-integer programming problems, making it accessible to users without deep optimization expertise.

**Strengths:**

- The paper provides a clear and compelling description of the motivation behind the research, along with a well-articulated explanation of the proposed method.
- It includes an extensive experimental comparison that effectively demonstrates the performance and advantages of the proposed approach relative to existing methods.

**Weaknesses:**

- In my experience with Operations Research, many papers focus on formulating new problems into linear programming or mixed-integer linear programming. These papers typically begin with a clear textual definition of the problem, followed by rigorous proofs, given the critical applications involved. However, I find that this work may not be particularly useful for practitioners in the field, as they still need to undertake similar formulation efforts themselves. The primary benefit seems to lie in providing students with examples for their coursework rather than advancing practical applications for researchers and professionals.

- In line 128, the quote from the textbook stating, "Once this formulation is done, solving the problem is ... (almost) technology," highlights a distinction from the problem formulation presented in this paper. The textbook emphasizes identifying the context of the problem, extracting the decision variables, and compiling all relevant real-world constraints, which cannot be adequately captured through mere textual descriptions.

**Questions:**

Could you clarify how the developed system will be utilized by domain scientists or in other potential use cases?

---

> ### Author Response · Authors · 2024-11-21
> **Response to Reviewer Utyj**
>
> *We appreciate the reviewer’s thoughtful evaluation and positive feedback.*
>
> ---
>
> ### [P1] Real-world utility
>
> Thank you for raising this important point. Our work focuses on autoformulation—an automated approach for translating natural language problem descriptions into diverse sets of mathematical and computational models. While our technical focus is on developing this capability, it is crucial to understand both its immediate practical value and long-term potential across different user groups.
>
> **Practical utility:** The benefits of autoformulation vary across different user groups: (1) For *domain experts without OR expertise* (e.g. business owners, healthcare administrators, engineers), it provides an accessible entry point to optimization techniques without requiring extensive OR training. (2) For *OR practitioners*, it holds the potential to streamline model development by automating routine formulation tasks, allowing greater focus on problem understanding and solution analysis. Much like how software engineers leverage coding LLMs to enhance productivity, autoformulation can serve as both an automation and ideation tool, facilitating rapid exploration of alternative formulations.
>
> **Our technical contributions:** While autoformulation technology is still emerging, our work makes several fundamental contributions at the intersection of OR and ML:
> * Formal definition of autoformulation as a search problem, with clear characterization of key challenges.
> * Novel framework combining MCTS with LLM-based hypothesis generation for systematic exploration under uncertainty.
> * Tailored pruning methods using SMT solvers to eliminate redundant formulations, enhancing search efficiency.
> * LLM-based evaluation approach for assessing formulation correctness through comparative evaluation.
>
> Our empirical results on IndustryOR, which contains challenging real-world problems, demonstrate significant improvements over existing approaches. Notably, it achieved a $10\%$ improvement over ORLM (an LLM finetuned for optimization modeling), validating the practical potential of our framework.
>
> **Looking ahead.** While we acknowledge the complexities of real-world optimization modeling (elaborated in [P2]), we see parallels with the evolution of LLM-based coding technologies, which progressed from simple function completion to full application development between 2021 and 2024. We anticipate autoformulation capabilities will follow a similar trajectory of rapid advancement, and our work aims to contribute to the foundations for these future developments.
>
> ---
>
> ### [P2] Real-world complexity
>
> Again, thank you for raising this important consideration. We acknowledge that real-world optimization modeling extends beyond the translation of problem descriptions into mathematical/computational models, and understanding these complexities helps identify which aspects can be effectively automated.
>
> Our autoformulation framework addresses a specific, yet crucial challenge: translating problem descriptions into mathematical and computational models. While this represents just one component of the broader modeling process, solving it alone is important for the reasons mentioned in [P1]. More broadly, the complete modeling process typically involves:
> 1. Information gathering from diverse stakeholders, often requiring multiple iterations and integration of implicit domain knowledge/conventions (i.e. 'tribal knowledge');
> 2. Handling complex problem characteristics such as stochasticity, time-varying dynamics, and large-scale variables;
> 3. Rigorous validation against real-world data, including sensitivity analysis of modeling assumptions and verification of edge cases;
> 4. Continuous communication between technical and business stakeholders to ensure practical utility.
>
> We would also like to use this opportunity to give an example where autoformulation can already play a large role in this process. The optimization problems in engineering systems are relatively 'well-defined'. A concrete example is wireless communications, where there are widely accepted requirements and metrics for measuring system performance (e.g. data rate, delay, energy consumption), commonly encountered decision variables, and the system under optimization is also relatively well understood (derived from physics or by construction of the system). In such domains, autoformulation could play a more comprehensive role in the modeling pipeline.
>
>
> **Actions taken:** We have included a discussion of real-world OR modeling complexities in `App H`.
>
> ---
> *We hope the reviewer’s concerns are addressed and they will consider updating their score. We welcome further discussions.*

---

> > ### Comment · Reviewer_Utyj · 2024-11-29
> > **reply**
> >
> > Thanks for addressing my concerns. I'm happy to improve my rating.
> >
> > Additionally, I would encourage the author to add a paragraph in related work on using MCTS and LLM to solve a diverse set of problems.

---

> > > ### Author Response · Authors · 2024-11-30
> > > **Thank you**
> > >
> > > Dear Reviewer **Utyj**,
> > >
> > > We sincerely appreciate your thorough engagement with our work and the concrete feedback that helped strengthen the paper. We are pleased that our responses addressed your comments and led to your recommendation for acceptance.
> > >
> > > Regarding your final suggestion, we have recently added an extended discussion of LLM+MCTS approaches and highlighted our novel contributions in App B. We will also include a condensed version of this analysis in the Related Work section.
> > >
> > >
> > > With thanks,
> > >
> > > The Authors of #10599

---

### Official Review · Reviewer_U3Rw · 2024-11-03

**Soundness:** 3
**Presentation:** 3
**Contribution:** 3
**Rating:** 6
**Confidence:** 4

**Summary:**

This paper introduces a framework to automate the creation of mathematical optimization models from natural language descriptions. This process is essential in operations research but traditionally requires significant expertise. There are three main challenges for autoformulation: defining a vast hypothesis space, navigating this space efficiently, and ensuring formulation correctness. To address these, the authors integrate LLMs within an MCTS framework, using LLMs to generate potential formulations and evaluate correctness. They also apply a pruning technique with SMT solvers to eliminate redundant formulations. Experiments on benchmarks for linear and mixed-integer programming show this approach is both accurate and efficient, potentially making optimization modeling more accessible.

**Strengths:**

1. Leveraging LLMs for both hypothesis generation and evaluation enables verification of partial problems.
2. Using MCTS to efficiently navigate the large hypothesis space, along with introducing the SMT pruning procedure, saves computational resources and focuses on unique, meaningful formulations.
3. Strong experimental results compared to baselines.

**Weaknesses:**

1. The same LLM is used for both hypothesis generation and verification, which introduces a high correlation between generation and evaluation stages. This could limit the objectivity and diversity of the verification.

2. Lacks a clear explanation of how the LLM is guided to explore the hypothesis space with sufficient diversity.

3. It would be beneficial to provide more insight into partial problem formulation at intermediate steps. Improved verification of partial models could enhance accuracy and reduce runtime for complex problems.


Minor issue: The font size in the figures is too small

**Questions:**

1. In Section 5.2, paragraph (2) ‘Getting Local Scores,’ how does the DFS expand the tree? Does it expand the child with the highest prior score first, or does it expand a random model generated by the LLM?

2. How should Figure 5 be interpreted? Section 5.4 doesn’t seem directly related.

3. How is exploration of the hypothesis space ensured? Figure 4 shows that pruning is effective, but does this imply that the LLM fails to generate diverse partial formulations?

---

> ### Author Response · Authors · 2024-11-21
> **Response to Reviewer U3Rw (1/2)**
>
> *We appreciate the reviewer’s detailed and thoughtful evaluation and positive feedback.*
>
> ---
>
> ### [P1] Correlation between model generation and evaluation
>
> Thank you for this important observation. We acknowledge that using the same LLM for both model generation and model evaluation could introduce correlations that potentially bias the autoformulation process.
>
> **Evaluation strategy.** Our method implements two key mechanisms to address this challenge. First, at the global level, we employ a composite evaluation strategy (`Eq 4`) that combines LLM-based assessment with solver feedback on optimality gap, providing complementary signals from independent sources. Additionally, for complete formulation evaluation, we utilize a ranking-based approach that compares candidates against baseline models rather than relying on absolute scoring, which helps mitigate self-reinforcing biases.
>
> **Empirical evidence.** Our empirical analysis supports the effectiveness of this approach. The ablation study in `Sec 5.2` reveals two key findings:
> 1. Ranking-based scoring of partial formulations significantly outperforms uniform scoring baselines (`Fig 3`), demonstrating the evaluation protocol's capacity to meaningfully differentiate between candidate expansions.
> 2. Comparative evaluation of complete formulations shows strong correlation ($\rho=0.48, p<0.001$) with ground-truth correctness, indicating robust evaluation despite using the same LLM.
>
> **Future direction.** This challenge reflects a broader open question in LLM research. Recent works have offered tentative evidence that LLMs have the potential to evaluate their own generations and iteratively improve [1, 2]. Our framework could be extended through dedicated generation and evaluation models, fine-tuned evaluators trained on expert feedback, or debiasing techniques using ensemble/mixture-based approaches.
>
> **Actions taken:** We have included a detailed discussion of these considerations and future research directions in `L515`.
>
>
> [1] Shinn, N., Cassano, F., Gopinath, A., Narasimhan, K. and Yao, S., 2024. Reflexion: Language agents with verbal reinforcement learning. Advances in Neural Information Processing Systems, 36.
>
> [2] Madaan, A., Tandon, N., Gupta, P., Hallinan, S., Gao, L., Wiegreffe, S., Alon, U., Dziri, N., Prabhumoye, S., Yang, Y. and Gupta, S., 2024. Self-refine: Iterative refinement with self-feedback. Advances in Neural Information Processing Systems, 36.
>
> ---
>
> ### [P2] Diversity in exploration
>
> We appreciate this question. As identified in `Sec 2.1`, efficient exploration of the vast hypothesis space is a key challenge in autoformulation that our method aims to explicitly address.
>
> **Encouraging diversity.** Our method incorporates multiple complementary mechanisms to ensure diverse exploration:
> 1. **Hierarchical decomposition** structures the search across four levels (variables, objective function, equality, and inequality constraints), enabling focused exploration within each decomposed component rather than searching for entire formulations at once.
> 2. **MCTS framework** with UCT scoring dynamically balances exploration-exploitation, adaptively guiding the search towards promising or unexplored regions of the hypothesis space.
> 3. **SMT-based pruning** removes trivially equivalent formulations, ensuring computational resources are devoted primarily to exploring functionally distinct solutions.
>
> **Experimental results.** The efficacy of our method design is evidenced by our empirical results.
> * `Fig 4` demonstrates the diversity of generated formulations at each hierarchical level, particularly highlighting how SMT pruning promotes exploration of functionally distinct paths while preventing exponential growth in redundant search efforts.
> * Additionally, `Fig 5` (further discussed in subsequent responses) illustrates that additional MCTS iterations consistently uncover new, correct formulations, confirming effective exploration of the solution space rather than redundant sampling or search.
>
> **Meaningful diversity.** A core premise of our work is leveraging LLMs as dynamic hypothesis generators that assign probability mass over plausible mathematical formulations. While our experimental results validate this assumption by demonstrating LLMs' ability to generate diverse formulations, the search process faces a key challenge: trivial variations due to mathematical equivalences. This underscores the importance of our pruning mechanism in eliminating uninformative diversity and focusing the search on meaningful variations.

---

> > ### Author Response · Authors · 2024-11-21
> > **Response to Reviewer U3Rw (2/2)**
> >
> > ---
> >
> > ### [P3] Insights on partial formulation evaluations
> >
> >
> > Thank you for the concrete recommendation. In our existing analysis (`Fig 3`), we compare the performance of our method against an ablation that assigns uniform scores to all children nodes, highlighting the efficacy of partial evaluations to improve search.
> >
> > **Additional insights.** To obtain more insights related to partial formulation evaluation, we introduced a new set of results in `App D.5`, which we will summarize here in brief. We measure the correlation between partial formulation scores and ground-truth correctness (defined as the percentage of correct leaf nodes in its subtree). Our analysis revealed that partial evaluations increase in correlation with ground-truth correctness as depth increases. Intuitively, this reveals that deeper nodes contain more complete formulation information, enabling more accurate evaluation. Earlier components show weaker correlations due to interdependencies between modeling elements. For instance, evaluating decision variables in isolation is challenging without understanding their role in objectives and constraints. This increased uncertainty at earlier stages reflects the inherent difficulty in assessing partial formulations without full context.
> >
> > These findings underscore the importance of hierarchical search strategies that maintain diverse exploration paths, particularly in early stages where evaluation signals are weaker.
> >
> > **Action taken:** We included additional analysis on partial evaluation in `App D.5`.
> >
> >
> > ---
> >
> > ### Questions
> >
> > * **Feedback on figures.** Thank you for this suggestion! We have improved the figures to enhance readability.
> > * **Q1.** The DFS approach proceeds as follows: (1) samples 10 child nodes at each step, (2) applies SMT pruning to remove trivial equivalences, (3) ranks the remaining nodes, and (4) retains up to three highest-scoring children for further exploration. While the search traverses through the highest-scored child node, we note that for the ablation study (in `Fig 3`), the exploration order does not affect the results, as we analyze the complete tree.
> > * **Q2.** `Fig 5` illustrates the search evolution of our MCTS relative to the number of rollouts, highlighting our method's ability to benefit from additional exploration (more iterations) to discover more correct solutions. We have revised the manuscript to clarify this point.
> > * **Q3.** Please see our response [P2] above.
> >
> > ---
> >
> > *We hope the reviewer’s concerns are addressed and they will consider updating their score. We welcome further discussions.*

---

### Official Review · Reviewer_e8hb · 2024-11-09

**Soundness:** 3
**Presentation:** 3
**Contribution:** 3
**Rating:** 5
**Confidence:** 3

**Summary:**

This paper is along the popular direction of using LLM to automatically fomulate mathematical optimization problems. It introduces a formal definition of autoformulation and identifies three main challenages. To address these challenges, the authors propose a method using large language models (LLMs) within a Monte-Carlo Tree Search (MCTS) framework. LLMs are utilized both as hypothesis generators and evaluators, while MCTS incrementally explores the formulation space. Additionally, a pruning technique based on Satisfiability Modulo Theories (SMT) solvers is introduced to eliminate redundant, trivially equivalent formulations. The method was tested on benchmarks (NL4OPT and IndustryOR) and show superior results.

**Strengths:**

1. The approach to utilize LLM both as hypothesis generators and evaluators seems novel.
2. A pruning technique based on Satisfiability Modulo Theories (SMT) solvers is introduced to eliminate redundant, trivially equivalent formulations, which can improve the search efficiency in my understanding.

**Weaknesses:**

1. The concept of autoformulation defined in this paper is broad, including mathematical formulation and computational formulation, and optimality gap and computational efficiency. However, the main contribution of this paper is on mathematical formulation if I understand correctly. The rest in the proposed concept is not addressed in this paper.
2. I do not see why adding optimality gap and computational efficiency to the evaluation metric. They're solver dependent and is not the responsbility of formulation.
3. The authors defined Types I to Types III problems, but I do not see how they are related to the main contribution of this paper.
Overall, I feel like the concept proposed in this paper is too broad and distract readers from the main contribution of this paper.

**Questions:**

1. Can you justify why optimality gap and computational efficiency, which depends on solver, are also included in the evalutation metric?
2. What is the relationship of Types I to Types III problems with the rest of this paper?

---

> ### Author Response · Authors · 2024-11-21
> **Response to Reviewer e8hb (1/3)**
>
> *We appreciate the reviewer’s thoughtful evaluation and positive feedback.*
>
> ---
>
> ### [P1] Problem definition scope
>
> Thank you for this thoughtful comment. Please allow us to clarify several points about our problem definition and its relationship to our proposed method.
>
> **Generalized definition.** We introduced a generalized problem definition for autoformulation, as this is a relatively new (and promising) direction at the intersection of ML and optimization modeling. Our framework consists of three key components that together address the autoformulation objective (`Eq 2`).
>
> | **Component** | **Description** | **Our method** |
> |---|---|---|
> | Problem description $\rightarrow$ mathematical model ($p_\phi$) | Transformation of natural language requirements into formal mathematical formulation | MCTS-based search with LLMs as hypothesis generators and evaluators |
> | Mathematical model $\rightarrow$ computational model ($p_\psi$) | Transformation of mathematical formulation into solver-compatible code | Custom-developed deterministic parser |
> | Evaluation metrics ($Q$) | Measure of formulation quality, e.g. optimality gap, formulation correctness, and computational efficiency | Dual evaluation approach combining solver feedback (optimality) and LLM assessment of formulation correctness |
>
> **Value of general framework.** Our proposed method represents one possible instantiation of this broader framework, addressing all three components while focusing primarily on the first component, which we identified as the most challenging and pressing frontier. The generality and completeness of our problem definition serve several important purposes:
> 1. Clearly delineates the multiple challenges that need to be addressed in this emerging field
> 2. Facilitates systematic identification of research opportunities, such as developing LLM-based techniques for solver configuration or specialized models for evaluating formulation correctness
>
> We would appreciate any further feedback you may have on this point.
>
>
>
> ---

---

> ### Author Response · Authors · 2024-11-21
> **Response to Reviewer e8hb (2/3)**
>
> ---
> ### [P2] Optimality gap and computational efficiency
>
> We appreciate you raising this point, which allows us to highlight why optimality gap and computational efficiency are both formulation and solver dependent. In the table below, we describe the impact of both **formulation** and **solver choice** on **optimality gap** and **computational efficiency** for different types of optimization problems.
>
>
> | Problem type | Factor | Impact on optimality gap | Impact on computational efficiency |
> |---|---|---|---|
> | **Type I**   (Originally Convex) | **Formulation** | **Minimal impact:** any correct (equivalent) formulation can achieve global optimality | **Medium impact:** choice of variable representation (e.g. structure-preserving formulations) can improve solution time |
> |  | **Solver** | **Minimal impact:** most commercial solvers can achieve similar optimality gaps | **Medium impact:** specialized solvers for specific problem structures (LP, QP, SOCP) can be faster |
> | **Type II**   (Non-convex but Convexifiable) | **Formulation** | **High impact:** correct, convexified reformulation enables achievable global optimality | **High impact:** reformulation complexity affects solution time, generally convex reformulations are solved faster |
> |  | **Solver** | **Medium impact:** solver ability to handle reformulated structures affect solution quality | **High impact:** solver must efficiently handle the specific structure of reformulation |
> | **Type III**   (Non-convex requiring Relaxation) | **Formulation** | **High impact:** quality of relaxation directly affects optimality gap and bounds tightness | **Medium impact:** relaxation complexity affects solution time, involving trade-offs between relaxation tightness and computational efficiency |
> |  | **Solver** | **High impact:** solver abilities on relaxed problem affects solution quality | **Medium impact:** solving relaxed problems may require specialized solvers, although general purpose solvers are roughly comparable |
>
> **Key observations:**
> 1. For Type I problems, formulation mainly affects computational efficiency rather than optimality.
> 2. For Type II problems, correct reformulation is crucial for both optimality and efficiency.
> 3. For Type III problems, both formulation and solver choices significantly impact the trade-off between optimality and efficiency.
>
> **Empirical evidence:** To support this analysis empirically, we compare the impact of both formulation and solver choice (including 8 solvers) on optimality and solution time. These results are included in the revised `App G`, and we will briefly summarize them here:
> 1. **Impact of formulation:** We observed that convex reformulation results in high-quality solutions with different convex solvers consistently. In particular, a convex reformulation had a 10,000x smaller optimality gap and 1,000x faster solution time than a general-purpose solver on the non-convex original formulation.
> 2. **Impact of solver:** Using a specialized convex solver resulted in 1,000x faster solution time than a general-purpose solver.
>
> This empirically confirms the importance of both formulation and solver on optimality gap and computational efficiency.
>
> **Actions taken:** Thank you again for this comment, which has helped us improve the paper with further clarifications and discussions. We have updated `App G` with empirical analysis of the impact of formulation and solvers on solution quality. We have also included the table above into `App E`.
>
> ---

---

> > ### Author Response · Authors · 2024-11-21
> > **Response to Reviewer e8hb (3/3)**
> >
> > ---
> >
> > ### [P3] Type I-III problems
> >
> >
> > We appreciate this constructive feedback. For any autoformulation method, the nature of the optimization problem under consideration presents notably different challenges. We aimed to encapsulate these nuances in the categorization of problem types:
> >
> > **Method design implications.** Different problem types require distinct autoformulation strategies. For Type I problems, the main challenge is ensuring correct variable/constraint identification. For Type II problems, the autoformulator must recognize opportunities for equivalent convexified reformulations. For Type III problems, the autoformulator should identify appropriate relaxation strategies that trade-off optimality with efficiency.
> >
> > **Evaluation implications.** Problem types inform the metrics used to assess autoformulation quality. Evaluation of Type I problems will focus on formulation correctness. Type II introduces additional emphasis on convexity and optimality. Type III must evaluate relaxation quality and recovery effectiveness.
> >
> > **Small tweaks.** However, we agree that the current placement of the problem categorization in `Sec 2.2` can affect readability and flow. To improve our paper organization and maintain focus on our main contributions, we will move `Sec 2.2` to the appendix, while retaining a brief overview in the main text.
> >
> > **Actions taken:** We have moved `Sec 2.2` to the appendix, this reorganization will help readers focus on our primary contributions while retaining important conceptual foundations for future work in autoformulation.
> >
> > ---
> > ### Questions
> > 1. Please see our response in `[P2]`
> > 2. Please see our response in `[P3]`
> >
> > ---
> > *We hope the reviewer’s concerns are addressed and they will consider updating their score. We welcome further discussion.*

---

### Author Response · Authors · 2024-11-21
**Global Response**

_We thank the reviewers for their constructive comments and their commitment to the review process._

---

We are encouraged that reviewers found our methodology novel, particularly in "utilizing LLM both as hypothesis generators and evaluators" (**e8hb**, **U3Rw**), enabling crucial "verification of partial problems" (**U3Rw**). The integration of MCTS with LLMs was recognized for its specific improvements "to fit the optimization scenario" (**mbTn**), while our SMT-based pruning technique was highlighted for its ability to "eliminate redundant, trivially equivalent formulations" and "improve search efficiency" (**e8hb**, **U3Rw**).

Reviewers appreciated the paper's clarity, noting its "good writing, easy to follow" presentation (**mbTn**) with a "clear and compelling description of the motivation" (**Utyj**). The technical contribution was strengthened by "sufficient experimental details" (**mbTn**) and "extensive experimental comparison" (**Utyj**), demonstrating "strong experimental results compared to baselines" (**U3Rw**).

We have also taken the reviewers’ feedback into account and made the following key changes to improve the paper:

* Relocated `Sec 2.2 Categorization of Problem Difficulty` to `App E` to improve flow and presentation
* Extended analysis of partial formulation evaluations (`App D.5`)
* Included empirical investigation on the impact of formulation and solver configuration on optimality gap and solution time (`App G`)

We have also uploaded a revised manuscript, where these changes are highlighted in teal for easier identification. We sincerely thank the reviewers for their valuable feedback on strengthening our work and remain open to further suggestions.

---
With thanks,

The Authors of #10599

---

### Meta-Review · Area_Chair_mtQv · 2024-12-19

**Metareview:**

The paper considers a relevant problem of autoformalization but currently falls below par for ICLR conference because of following reasons:

- The paper uses a straightforward combination of existing ideas, each of which are widely studied in the literature now. For example, there is a lot of work on combining tree search with LLMs. This paper is mostly about applying existing techniques to a new problem. This is not necessarily bad because applying existing methods to new domains can be valuable, particularly when addressing real-world challenges but the current work primarily demonstrates these techniques on standard benchmarks. The broader claims about presenting a general framework lack enough technical validation.

- It would have been interesting to see any challenges that arise when applying these existing techniques to this problem and how they are addressed. For example, evaluating candidate formulations using LLMs is widely studied as “LLM-as-a-Judge”. It is presented as a gold standard solution but that is far from true.

- The paper has a disconnect between the problem formulation and actual implementation. Reviewer mbTn and e8hb have similar points in their review.

Please consider incorporating comments from the reviewers in the next cycle to improve the paper.

**Additional Comments On Reviewer Discussion:**

Reviewer mbTn and e8hb mentioned about between the problem formulation and actual implementation. Reviewers also had concerns about the claims made in the paper making more general contributions than the actual implemented ideas.

---

### Decision · Program_Chairs · 2025-01-22

Reject